# Geometric Trajectory Diffusion Models

**Jiaqi Han, Minkai Xu, Aaron Lou, Haotian Ye, Stefano Ermon**
Stanford University

## Abstract

Generative models have shown great promise in generating 3D geometric systems, which is a fundamental problem in many natural science domains such as molecule and protein design. However, existing approaches only operate on static structures, neglecting the fact that physical systems are always dynamic in nature. In this work, we propose geometric trajectory diffusion models (GeoTDM), the first diffusion model for modeling the temporal distribution of 3D geometric trajectories. Modeling such distribution is challenging as it requires capturing both the complex spatial interactions with physical symmetries and temporal correspondence encapsulated in the dynamics. We theoretically justify that diffusion models with equivariant temporal kernels can lead to density with desired symmetry, and develop a novel transition kernel leveraging SE(3)-equivariant spatial convolution and temporal attention. Furthermore, to induce an expressive trajectory distribution for conditional generation, we introduce a generalized learnable geometric prior into the forward diffusion process to enhance temporal conditioning. We conduct extensive experiments on both unconditional and conditional generation in various scenarios, including physical simulation, molecular dynamics, and pedestrian motion. Empirical results on a wide suite of metrics demonstrate that GeoTDM can generate realistic geometric trajectories with significantly higher quality.[1]

## 1 Introduction

Machine learning for geometric structures is a fundamental task in many natural science problems ranging from particle systems driven by physical laws [1, 26, 43, 2, 15] to molecular dynamics in biochemistry [22, 16, 45, 10]. Modeling such geometric data is challenging due to the physical symmetry constraint [56, 43], making it fundamentally different from common scalar non-geometric data such as images and text. With the recent progress of generative models, many works have been proposed in generating 3D geometric structures like small molecules [66, 42, 21, 64] and proteins [59, 23], showing great promise in solving the equilibrium states of complex systems.

Despite this success, these existing methods are limited to synthesizing static structures and neglect the fact that important real-world processes evolve through time. For example, molecules and proteins are not static but always varying with molecular dynamics, which plays a vital role in analyzing possible binding activities [8, 20]. In this paper, we aim to study the generative modeling of geometric trajectories with the additional temporal dimension. While this problem is more practical and important, it is highly non-trivial with several significant challenges. First, geometric dynamics in 3D ubiquitously preserve physical symmetry. With global translation or rotation applied to a trajectory of molecular dynamics, the entire trajectory still describes the same dynamics and the generative model should estimate the same likelihood. Second, trajectories inherently contain the correspondence between frames in different timesteps, requiring generative models to hold a high capacity for capturing the temporal correlations. Last, moving from a single structure to a trajectory

---

[1]Correspondence to Jiaqi Han: jiaqihan@stanford.edu. Code is available at https://github.com/hanjq17/GeoTDM.

38th Conference on Neural Information Processing Systems (NeurIPS 2024).

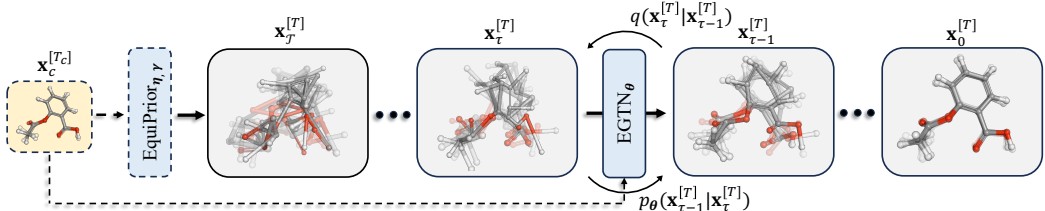

Figure 1: Overview of GeoTDM. The forward diffusion $q$ gradually perturbs the input while the reverse process $p_{\boldsymbol{\theta}}$, parameterized by EGTN, denoises samples from the prior. The condition $\mathbf{x}_c^{[T_c]}$, if available, is leveraged to construct the equivariant prior and as a conditioning signal in EGTN.

composed of multiple ones, the distribution we are interested in becomes much higher-dimensional and more diverse, considering both the initial conditions as well as potential uncertainties injected along the evolution of dynamics.

To this end, we propose geometric trajectory diffusion models (GeoTDM), a principled method for modeling the temporal distribution of geometric trajectories through diffusion models [49, 52, 53, 18], the state-of-the-art generative model on various domains such as images [6], videos [19], and molecules [64]. Our key innovation lies in designing an equivariant temporal diffusion over geometric trajectories, with the reverse process parameterized by equivariant transition kernels, ensuring the desired physical symmetry of the generated trajectory. To better excavate the complex spatial interactions and temporal correlations, we develop a novel temporal denoising network, where we stack equivariant spatial convolution and temporal attention. Our developments not only guarantee the desirable physical symmetry of the trajectories, but also capture the complex spatial and temporal correspondence encapsulated in the dynamics of geometric systems. Moreover, by leveraging generative modeling, GeoTDM enjoys high versatility in generating diverse yet high-quality geometric trajectories from scratch, performing interpolation and extrapolation, and optimizing noisy trajectory, all under the proposed diffusion framework.

In summary, we make the following contributions: **1.** We present GeoTDM, a novel temporal diffusion model for generating geometric trajectories. We design the diffusion process to meet the critical equivariance in modeling both unconditional and conditional distributions over geometric trajectories. Notably, we also propose a conditional learnable equivariant prior for enhanced flexibility in temporal conditioning. **2.** To fulfill the equivariance of the denoising network, we introduce EGTN, a graph neural network that operates on geometric trajectories, which also permits conditioning upon a given trajectory using equivariant cross-attention, making it suitable to serve as the backbone for GeoTDM. **3.** We evaluate our GeoTDM on both unconditional and conditional trajectory generation tasks including particle simulation, molecular dynamics, and pedestrian trajectory prediction. GeoTDM can consistently outperform existing approaches on various metrics, with up to 56.7% lower prediction score for unconditional generation and 16.8% lower forecasting error for conditional generation on molecular dynamics simulation. We also show GeoTDM successfully performs several additional applications, such as temporal interpolation and trajectory optimization.

## 2 Related Work

**Trajectory modeling for geometric systems.** Modeling the dynamics of geometric data is challenging since one must capture the interactions between multiple objects. Graph neural networks [11] have emerged as a natural tool to tackle this complexity [26, 41]. Subsequent works [43, 7, 2, 63] discovered equivariance as a critical factor for promoting model generalization. Among these efforts, Radial Fields [27] and EGNN [43] work with equivariant operations between scalars and vectors, while TFN [56] and SE(3)-Transformer [9] generalize to high-order spherical tensors. While considerable progress has been made, they only conduct (time) frame-to-frame prediction, which is subject to error accumulation when performing roll-out inference. Recently, EqMotion [62] approached the problem by learning to predict trajectories. By comparison, our GeoTDM leverages a generative modeling framework, which enables a wider range of tasks such as generation and interpolation.

**Generative models in geometric domain.** There is growing interest in developing generative models for geometric data, *e.g.* molecule generation [42, 66, 21, 64], protein generation [59, 24, 69], and

antibody design [30]. Recently, diffusion-based models [21, 64] have been shown to yield superior performance compared to flow-based [42] and VAE-based [65] approaches in many of these tasks. Despite these fruitful achievements, most existing works only produce a snapshot of the geometric system, *e.g.*, a molecule in 3D space, whereas our GeoTDM generalizes to generating a trajectory with multiple frames, *e.g.*, an MD trajectory in 3D. DiffMD [60] specifically tackles MD modeling using Markovian assumption, while GeoTDM directly captures the joint distribution of all frames along the entire trajectory.

**Temporal diffusion models.** Diffusion models have been recently adapted to handle the natural temporality of data in tasks such as video generation [19, 58, 17], time series forecasting [37, 54], PDE simulation [38], human motion synthesis [55, 71] and pedestrain trajectory forecasting [12]. Distinct from these works, GeoTDM models the temporal evolution of geometric data represented as a geometric graph and maintains the aforementioned vital equivariance constraint.

## 3 Preliminaries

**Diffusion models.** Diffusion models [49, 18, 52, 53] are a type of latent variable generative model that feature a Markovian forward diffusion process and reverse denoising process. The forward process progressively perturbs the input $\mathbf{x}_0$ (*e.g.*, image pixels or molecule coordinates) over $\mathcal{T}$ steps using a Gaussian transition kernel $q(\mathbf{x}_\tau|\mathbf{x}_{\tau-1}) = \mathcal{N}(\mathbf{x}_\tau; \sqrt{1-\beta_\tau}\mathbf{x}_{\tau-1}, \beta_\tau\mathbf{I})$. Here, $\{\mathbf{x}_\tau\}_{\tau=1}^{\mathcal{T}}$ are latent variables with the same dimension as the input and $\beta_\tau$ are predefined using the noise schedule such that $\mathbf{x}_{\mathcal{T}}$ is close to being distributed as $\mathcal{N}(\mathbf{0}, \mathbf{I})$. The reverse process maps back from the prior distribution with $p(\mathbf{x}_{\mathcal{T}}) = \mathcal{N}(\mathbf{0}, \mathbf{I})$ using the kernel $p_{\boldsymbol{\theta}}(\mathbf{x}_{\tau-1}|\mathbf{x}_\tau) = \mathcal{N}(\mathbf{x}_{\tau-1}; \boldsymbol{\mu}_{\boldsymbol{\theta}}(\mathbf{x}_\tau, \tau), \sigma_\tau^2\mathbf{I})$, where the variances $\sigma_\tau^2$ are usually fixed and the mean $\boldsymbol{\mu}_{\boldsymbol{\theta}}$ is parameterized by a neural network with parameters $\boldsymbol{\theta}$. The model is trained by optimizing the variational lower bound, defined as $\mathcal{L}_{\text{vlb}} = -\log p_{\boldsymbol{\theta}}(\mathbf{x}_0|\mathbf{x}_1) + D_{\text{KL}}(q(\mathbf{x}_{\mathcal{T}}|\mathbf{x}_0)\|p(\mathbf{x}_{\mathcal{T}})) + \sum_{\tau=2}^{\mathcal{T}-1} D_{\text{KL}}(q(\mathbf{x}_{\tau-1}|\mathbf{x}_\tau, \mathbf{x}_0)\|p_{\boldsymbol{\theta}}(\mathbf{x}_{\tau-1}|\mathbf{x}_\tau))$. For training stability, [52, 18] suggest the noise-prediction objective:

$$\mathcal{L}_{\text{simple}} := \mathbb{E}_{\mathbf{x}_0, \boldsymbol{\epsilon} \sim \mathcal{N}(\mathbf{0}, \mathbf{I}), \tau} \lambda(\tau) \left[ \|\boldsymbol{\epsilon} - \boldsymbol{\epsilon}_{\boldsymbol{\theta}}(\mathbf{x}_\tau, \tau)\|^2 \right], \tag{1}$$

where $\mathbf{x}_0 \sim p_{\text{data}}$, $\tau \sim \text{Unif}(1, \mathcal{T})$, the weighting factors $\lambda(\tau)$ are typically set to 1 to promote sample quality, $\mathbf{x}_\tau = \sqrt{\bar{\alpha}_\tau}\mathbf{x}_0 + \sqrt{1-\bar{\alpha}_\tau}\boldsymbol{\epsilon}$ with $\bar{\alpha}_\tau := \prod_{s=1}^{\tau} \alpha_s = \prod_{s=1}^{\tau}(1-\beta_s)$, and $\boldsymbol{\epsilon}_{\boldsymbol{\theta}}$ is a specific parameterization of the mean satisfying $\boldsymbol{\mu}_{\boldsymbol{\theta}}(\mathbf{x}_\tau, \tau) = \frac{1}{\sqrt{\alpha_\tau}}(\mathbf{x}_\tau - \frac{\beta_\tau}{\sqrt{1-\bar{\alpha}_\tau}}\boldsymbol{\epsilon}_{\boldsymbol{\theta}}(\mathbf{x}_\tau, \tau))$.

**Equivariance.** *Functions.* A function $f$ is equivariant *w.r.t* a group $G$ if $f(g \cdot \mathbf{x}) = g \cdot f(\mathbf{x}), \forall g \in G$. Furthermore, $f$ is invariant if $f(g \cdot \mathbf{x}) = f(\mathbf{x}), \forall g \in G$ [46]. Here we focus on the group SE(3) consisting of all 3D rotations and translations[2]. Each group element $g \in$ SE(3) can be represented by a rotation matrix $\mathbf{R}$ and a translation $\mathbf{r} \in \mathbb{R}^3$. For geometrc graph with node features $\mathbf{h}$ and coordinates $\mathbf{x}$, if $\mathbf{h}', \mathbf{x}' = f(\mathbf{h}, \mathbf{x})$, we expect $\mathbf{h}', \mathbf{R}\mathbf{x}' + \mathbf{r} = f(\mathbf{h}, \mathbf{R}\mathbf{x} + \mathbf{r})$[3], *i.e.*, the output node features are invariant while the updated coordinates are equivariant. *Distributions.* We call a density $p(\mathbf{x})$ invariant *w.r.t.* a group $G$ if $p(g \cdot \mathbf{x}) = p(\mathbf{x}), \forall g \in G$. Intuitively, geometries that are rotationally and translationally equivalent should share the same density, since they all refer to the same structure. A conditional distribution $p(\mathbf{x}|\mathbf{y})$ is equivariant if $p(g \cdot \mathbf{x}|g \cdot \mathbf{y}) = p(\mathbf{x}|\mathbf{y}), \forall g \in G$. Such a property is important in cases where the target distribution is conditioned on some given structures: if the observed geometry is rotated/translated, the target distribution should also rotate/translate accordingly.

**Geometric trajectories and the distributions.** We represent a geometric trajectory as $(\mathbf{x}^{[T]}, \mathbf{h}, \mathcal{E})$, where $\mathbf{x}^{[T]} := \left[\mathbf{x}^{(0)}, \mathbf{x}^{(1)}, \cdots, \mathbf{x}^{(T-1)}\right] \in \mathbb{R}^{T \times N \times D_\mathbf{x}}$ is the sequence of temporal geometric coordinates, $\mathbf{h} \in \mathbb{R}^{N \times D_\mathbf{h}}$ is the node feature, and $\mathcal{E}$ is the set of edges representing the connectivity of the geometric graph. $T$ is the number of time steps and $D_\mathbf{x}, D_\mathbf{h}$ refers to the dimension of the coordinate and node feature respectively, with $D_\mathbf{x}$ normally being 2 or 3 depending on the input data. In this work, we are interested in modeling the distribution of geometric trajectories given the configuration of the geometric graph, *i.e.*, $p(\mathbf{x}^{[T]}|\mathbf{h}, \mathcal{E})$.

**Conditioning.** Some applications like trajectory forecasting can be viewed as conditional generative tasks, where we seek to model the distribution of trajectories conditioning on certain observed timesteps, *i.e.*, $p(\mathbf{x}^{[T]}|\mathbf{x}_c^{[T_c]}, \mathbf{h}, \mathcal{E})$ where $\mathbf{x}_c^{[T_c]} \in \mathbb{R}^{T_c \times N \times D_\mathbf{x}}$ is the provided trajectory in length $T_c$.

---

[2]The analyses in this paper also hold for the general $n$D case, *e.g.*, 2D.

[3]Following convention we use the notation $\mathbf{R}\mathbf{x}$ to denote $\mathbf{x}\mathbf{R}^\top$.

**Equivariance for geometric trajectories.** Since the dynamics must be invariant to rotation or translation, the distribution of the geometric trajectories should also preserve such symmetry. This is formalized by the following invariance constraint:

$$p(\mathbf{x}^{[T]} \mid \mathbf{h}, \mathcal{E}) = p(g \cdot \mathbf{x}^{[T]} \mid \mathbf{h}, \mathcal{E}), \forall g \in \mathrm{SE}(3). \tag{2}$$

where $g \cdot \mathbf{x}^{[T]} := [\mathbf{R}\mathbf{x}^{(0)} + \mathbf{r}, \cdots, \mathbf{R}\mathbf{x}^{(T-1)} + \mathbf{r}]$. The conditional case should instead preserve:

$$p(\mathbf{x}^{[T]} | \mathbf{x}_c^{[T_c]}, \mathbf{h}, \mathcal{E}) = p(g \cdot \mathbf{x}^{[T]} | g \cdot \mathbf{x}_c^{[T_c]}, \mathbf{h}, \mathcal{E}), \tag{3}$$

for all $g \in \mathrm{SE}(3)$[4]. Intuitively, if the given trajectory is rotated and/or translated, the distribution of the future trajectory should also rotate and/or translate by exactly the same amount. For simplicity, we omit writing the conditions $\mathbf{h}$ and $\mathcal{E}$ henceforth when describing the distributions of trajectories.

## 4 Geometric Trajectory Diffusion Models

In this section, we introduce the machinery of GeoTDM. We first present Equivariant Geometric Trajectory Network (EGTN) in § 4.1, a general purpose backbone operating on geometric trajectories while ensuring equivariance. We then present GeoTDM in § 4.2 for both unconditional and conditional generation using EGTN as the denoising network.

### 4.1 Equivariant Geometric Trajectory Network

Our proposed Equivariant Geometric Trajectory Network (EGTN) is constructed by stacking equivariant spatial aggregation layers and temporal attention layers in an alternated manner, drawing inspirations from spatio-temporal GNNs [70, 61]. In particular, spatial layers characterize the structural interactions within the system and temporal layers model the temporal dependencies along the trajectory. For spatial aggregation, we employ the Equivariant Graph Convolution Layer (EGCL) [43],

$$\mathbf{x}'^{(t)}, \mathbf{h}'^{(t)} = \mathrm{EGCL}(\mathbf{x}^{(t)}, \mathbf{h}^{(t)}, \mathcal{E}), \forall t \in [T]. \tag{4}$$

The equivariant message passing is conducted independently for each frame $t \in [T] := \{0, 1, \cdots, T - 1\}$, with the goal of passing and fusing the geometric information based on the structure of the graph for each time step. Following such layer, we further develop a temporal layer equipped by self-attention, which has exhibited great promise for sequence modeling [57], to capture the temporal correlations encapsulated in the dynamics. We first compute Eqs. 5-6, where $\mathbf{q}^{(t)}, \mathbf{k}^{(t,s)}, \mathbf{v}^{(t,s)}$ are the query, key, and value, respectively. In detail, $\mathbf{q}^{(t)} = \varphi_{\mathbf{q}}(\mathbf{h}^{(t)})$, $\mathbf{k}^{(t,s)} = \varphi_{\mathbf{k}}(\mathbf{h}^{(s)}) + \psi(t - s)$, and $\mathbf{v}^{(t,s)} = \varphi_{\mathbf{v}}(\mathbf{h}^{(s)}) + \psi(t - s)$, with $\psi(t - s)$ being the sinusoidal encoding [57] of the temporal displacement $t - s$, akin to the relative positional encoding [47]. Incorporating such information is

$$a^{(t,s)} = \frac{\exp\left(\mathbf{q}^{(t)\top}\mathbf{k}^{(t,s)}\right)}{\sum_{u \in [T]} \exp\left(\mathbf{q}^{(t)\top}\mathbf{k}^{(t,u)}\right)}, \tag{5}$$

$$\mathbf{h}'^{(t)} = \mathbf{h}^{(t)} + \sum_{s \in [T]} a^{(t,s)}\mathbf{v}^{(t,s)}, \tag{6}$$

crucial since the model is supposed to distinguish different time spans between two frames on the trajectory. Moreover, compared with directly encoding the absolute time step, our design is beneficial in that it ensures the temporal shift invariance of physical processes. The update of coordinates reuse the attention coefficients $a^{(t,s)}$ and the values $\mathbf{v}^{(t,s)}$,

$$\mathbf{x}'^{(t)} = \mathbf{x}^{(t)} + \sum_{s \in [T]} a^{(t,s)}\varphi_{\mathbf{x}}(\mathbf{v}^{(t,s)})(\mathbf{x}^{(t)} - \mathbf{x}^{(s)}), \tag{7}$$

where $\varphi_{\mathbf{x}}$ is an MLP that outputs a scalar to preserve rotation equivariance. The entire network $f_{\mathrm{EGTN}}$, with schematic depicted in Fig. 4, is constructed by alternating spatial and temporal layers, enjoying equivariance as desired (proof in Appendix A.4):

**Theorem 4.1** (SE(3)-equivariance). *Let* $\mathbf{x}'^{[T]}, \mathbf{h}'^{[T]} = f_{\mathrm{EGTN}}\left(\mathbf{x}^{[T]}, \mathbf{h}^{[T]}, \mathcal{E}\right)$. *Then we have* $g \cdot \mathbf{x}'^{[T]}, \mathbf{h}'^{[T]} = f_{\mathrm{EGTN}}\left(g \cdot \mathbf{x}^{[T]}, \mathbf{h}^{[T]}, \mathcal{E}\right), \forall g \in SE(3)$.

**Geometric conditioning.** In certain tasks like trajectory forecasting, we are additionally provided with some partially observed trajectories as side input. In order to leverage their geometric information,

---

[4]Technically, such a condition is impossible since SE(3) is noncompact, but we show that zero-centering the trajectories and enforcing SO(3)-invariance is equivalent.

we augment the unconditional EGTN with *equivariant cross-attention*, a conditioning technique tailored for geometric trajectories, and more importantly, guaranteeing the crucial equivariance in Theorem 4.1. In principle, our equivariant cross-attention resembles Eqs. 5-7, but instead computes the attention between the conditioning trajectory $\mathbf{x}_c^{[T_c]}$ and the target $\mathbf{x}^{[T]}$. In detail, the attention coefficients are recomputed as $\mathbf{a}^{(t,s)} = \frac{\exp\left(\mathbf{q}^{(t)\top}\mathbf{k}^{(t,s)}\right)}{\sum_{u \in [T_c] \cup [T]} \exp\left(\mathbf{q}^{(t)\top}\mathbf{k}^{(t,u)}\right)}$. The updated node feature $\mathbf{h}'^{(t)}$ and coordinate $\mathbf{x}'^{(t)}$ in Eqs. 6-7 are further renewed by the cross-attention terms, yielding $\mathbf{h}''^{(t)} = \mathbf{h}'^{(t)} + \sum_{s \in [T_c]} \mathbf{a}^{(t,s)} \mathbf{v}^{(t,s)}$ and $\mathbf{x}''^{(t)} = \mathbf{x}'^{(t)} + \sum_{s \in [T_c]} \mathbf{a}^{(t,s)} \varphi_{\mathbf{x}}(\mathbf{v}^{(t,s)})(\mathbf{x}^{(t)} - \mathbf{x}_c^{(s)})$.

## 4.2 Geometric Trajectory Diffusion Models

### 4.2.1 Unconditional Generation

For unconditional generation, we seek to model the trajectory distribution subject to the SE(3)-invariance (Eq. 2). To design a diffusion with the reverse marginal conforming to the invariance, we impose certain constraints to the prior and transition kernel, as depicted in the following theorem.

**Theorem 4.2.** *If the prior $p_{\mathcal{T}}(\mathbf{x}_{\mathcal{T}}^{[T]})$ is SE(3)-invariant, the transition kernels $p_{\tau-1}(\mathbf{x}_{\tau-1}^{[T]} \mid \mathbf{x}_{\tau}^{[T]}), \forall \tau \in \{1, \cdots, \mathcal{T}\}$ are SE(3)-equivariant, then the marginal $p_{\tau}(\mathbf{x}_{\tau}^{[T]})$ at any step $\tau \in \{0, \cdots, \mathcal{T}\}$ is also SE(3)-invariant.*

**Prior in the translation-invariant subspace.** Unfortunately, there is no properly normalized distribution *w.r.t.* Lebesgue measure on the ambient space $\mathcal{X} := \mathbb{R}^{T \times N \times D}$ that permits translation-invariance [43]. We instead build the prior on a translation-invariant subspace $\mathcal{X}_{\mathbf{P}} \subset \mathcal{X}$ induced by a linear transformation $\mathbf{P} \in \mathcal{X} \times \mathcal{X}$ with $\mathrm{rank}(\mathbf{P}) = (TN-1)D$ [36]. Specifically, we choose the prior to be the projection of the Gaussian $\mathcal{N}(\mathbf{0}, \mathbf{I})$ in $\mathcal{X}$ to $\mathcal{X}_{\mathbf{P}}$ by $\mathbf{P} = \mathbf{I}_D \otimes \left(\mathbf{I}_{TN} - \frac{1}{TN}\mathbf{1}_{TN}\mathbf{1}_{TN}^{\top}\right)$, which corresponds to the function $P(\mathbf{x}^{[T]}) = \mathbf{x}^{[T]} - \frac{1}{T}\sum_{t=0}^{T-1}\mathrm{CoM}(\mathbf{x}^{(t)})$, with $\mathrm{CoM}(\mathbf{x}^{(t)}) = \frac{1}{N}\sum_{i=1}^{N}\mathbf{x}_i^{(t)}$ being the center-of-mass (CoM) of the system at time $t$. We denote $\tilde{\mathbf{x}} := P(\mathbf{x})$. Then the resulting distribution is a restricted Gaussian (denoted $\tilde{\mathcal{N}}(\mathbf{0}, \mathbf{I})$) with the variables supported only on the subspace (see App. A.1), and more importantly, is still isotropic and thus SO(3)-invariant. To sample from the prior, one can alternatively sample $\mathbf{x}^{[T]} \sim \mathcal{N}(\mathbf{0}, \mathbf{I}) \in \mathcal{X}$ and then project it to the subspace to obtain the final sample $\tilde{\mathbf{x}}^{[T]} = P(\mathbf{x}^{[T]}) \in \mathcal{X}_{\mathbf{P}}$.

**Transition kernel.** To be consistent with the prior, we also parameterize the transition kernel in the subspace $\mathcal{X}_{\mathbf{P}}$, given by $p_{\theta}(\tilde{\mathbf{x}}_{\tau-1}^{[T]} \mid \tilde{\mathbf{x}}_{\tau}^{[T]}) = \tilde{\mathcal{N}}(\tilde{\boldsymbol{\mu}}_{\theta}(\tilde{\mathbf{x}}_{\tau}^{[T]}, \tau), \sigma_{\tau}^2\mathbf{I})$. In this way, if the mean function $\tilde{\boldsymbol{\mu}}_{\theta}(\cdot)$ is SO(3)-equivariant, then the transition kernel is also guaranteed SO(3)-equivariance. As suggested by [18], we re-parameterize $\tilde{\boldsymbol{\mu}}_{\theta}(\tilde{\mathbf{x}}_{\tau}^{[T]}, \tau) = \frac{1}{\sqrt{\alpha_{\tau}}}\left(\tilde{\mathbf{x}}_{\tau}^{[T]} - \frac{\beta_{\tau}}{\sqrt{1-\bar{\alpha}_{\tau}}}\tilde{\boldsymbol{\epsilon}}_{\theta}(\mathbf{x}_{\tau}^{[T]}, \tau)\right)$, where $\tilde{\boldsymbol{\epsilon}}_{\theta} = P \circ \mathbf{f}_{\theta}$, with $\mathbf{f}_{\theta}$ being an SO(3)-equivariant adaptation of our proposed EGTN, fulfilled by subtracting the input coordinates from the output for translation invariance. The diffusion step $\tau$ is transformed via time embedding and concatenated to the invariant node features $\mathbf{h}^{[T]}$ in the input.

**Training and inference.** We optimize the VLB for training, which, interestingly, still has a surrogate in the noise-prediction form when specifying the factors $\lambda(\tau)$ as 1 (proof in App. A.1):

$$\mathcal{L}_{\mathrm{uncond}} := \mathbb{E}_{\mathbf{x}_0^{[T]}, \tilde{\boldsymbol{\epsilon}} \sim \tilde{\mathcal{N}}(\mathbf{0}, \mathbf{I}), \tau \sim \mathrm{Unif}(1, \mathcal{T})}\left[\|\tilde{\boldsymbol{\epsilon}} - \tilde{\boldsymbol{\epsilon}}_{\theta}(\tilde{\mathbf{x}}_{\tau}^{[T]}, \tau)\|^2\right]. \tag{8}$$

The inference process is similar to [18] but with additional applications of $P$ in intermediate steps to keep all samples in the subspace $\mathcal{X}_{\mathbf{P}}$. Details are in Alg. 1 and 2.

### 4.2.2 Conditional Generation

Distinct from the unconditional generation, in the conditional scenario the target distribution should instead be SE(3)-equivariant *w.r.t.* the given frames, as elucidated in Eq. 3. The following theorem describes the constraints to consider when designing the prior and transition kernel.

**Theorem 4.3.** *If the prior $p_{\mathcal{T}}(\mathbf{x}_{\mathcal{T}}^{[T]}|\mathbf{x}_c^{[T_c]})$ is SE(3)-equivariant, the transition kernels $p_{\tau-1}(\mathbf{x}_{\tau-1}^{[T]}|\mathbf{x}_{\tau}^{[T]}, \mathbf{x}_c^{[T_c]}), \forall \tau \in \{1, \cdots, \mathcal{T}\}$ are SE(3)-equivariant, the marginal[5] $p_{\tau}(\mathbf{x}_{\tau}^{[T]}|\mathbf{x}_c^{[T_c]}), \forall \tau \in \{0, \cdots, \mathcal{T}\}$ is SE(3)-equivariant.*

---

[5] Marginal refers to marginalizing the intermediate states in reverse process while still conditioning on $\mathbf{x}_c^{[T_c]}$.

**Flexible equivariant prior.** There are in general many valid choices for the prior while satisfying SE(3)-equivariance. We provide a guidance on distinguishing feasible designs when using Gaussian-based prior in the proposition below.

**Proposition 4.4.** $\mathcal{N}(\boldsymbol{\mu}(\mathbf{x}_c^{[T_c]}), \mathbf{I})$ *is SE(3)-equivariant w.r.t.* $\mathbf{x}_c^{[T_c]}$ *if* $\boldsymbol{\mu}(\mathbf{x}_c^{[T_c]})$ *is SE(3)-equivariant.*

Proof is in App. A.2. Notably, the mean of the prior $\mathbf{x}_r^{[T]} := \boldsymbol{\mu}(\mathbf{x}_c^{[T_c]})$ naturally serves as an anchor to transit the geometric information in the provided trajectory to the target distribution we seek to model. For instance, one can choose it as a linear combination of the CoMs of the given frames, *i.e.*, $\mathbf{x}_r^{[T]} = \mathbf{1}_{T \times N} \otimes \sum_{s \in [T_c]} w^{(s)} \bar{\mathbf{x}}_c^{(s)}$, where $\sum_{s \in [T_c]} w^{(s)} = 1$ are fixed parameters determined *a priori* [21, 13]. However, this choice does not leverage temporal consistency of the trajectory and incurs extra effort in optimization, since the model needs to learn to reconstruct the complex structures from points all located at the CoM. In contrast, we propose the following instantiation:

$$\mathbf{x}_r^{(t)} = \sum_{s \in [T_c]} \mathbf{w}^{(t,s)} \hat{\mathbf{x}}_c^{(s)}, \quad \text{s.t.} \sum_{s \in [T_c]} \mathbf{w}^{(t,s)} = \mathbf{1}, \tag{9}$$

for all $t \in [T]$, where each $\mathbf{x}_r^{(t)}$ is a point-wise linear combination of $\hat{\mathbf{x}}_c^{(s)}$, an SE(3)-equivariant transformation of the conditioning frames, with $\mathbf{w}^{(t,s)} \in \mathbb{R}^N$ being the weights. We first obtain $\hat{\mathbf{x}}_c^{[T_c]}, \hat{\mathbf{h}}_c^{[T_c]} = \mathbf{f}_\eta(\mathbf{x}_c^{[T_c]}, \mathbf{h}_c^{[T_c]})$ where $\mathbf{f}_\eta$ is a lightweight two layer EGTN that aims to synthesize the conditional information. The $\mathbf{w}^{(t,s)}$ is then derived as,

$$\mathbf{W}_{t,s} = [\boldsymbol{\gamma} \otimes \hat{\mathbf{h}}_c^{[T_c]}]_{t,s} \in \mathbb{R}^N, \tag{10}$$

$$\mathbf{w}^{(t,s)} = \begin{cases} \mathbf{W}_{t,s} & s < T_c - 1, \\ \mathbf{1}_N - \sum_{s=0}^{T_c-2} \mathbf{W}_{t,s} & s = T_c - 1. \end{cases} \tag{11}$$

Here $\boldsymbol{\gamma} \in \mathbb{R}^T$ are learnable parameters, and $\mathbf{w}^{(t,s)}$ is parameterized such that it has a sum of $\mathbf{1}_N$ when $s$ goes through $[T_c]$ to satisfy the constraint in Eq. 9 for translation equivariance. Interestingly, as we formally illustrated in Theorem A.4, our parameterization of the prior theoretically *subsumes* the CoM-based priors [21, 13] and the fixed point-wise priors when $\boldsymbol{\gamma}$, $\hat{\mathbf{h}}_c^{[T_c]}$, and $\hat{\mathbf{x}}_c^{[T_c]}$ reduce to specific values. Such theoretical result underscores the benefit of our design since it permits the model to dynamically update the prior, leading to better optimization. The parameters $\boldsymbol{\eta}$ and $\boldsymbol{\gamma}$ are updated during training with gradients coming from optimizing the variational lower bound.

**Transition kernel.** We need to modify the forward and reverse process such that they both match the proposed prior. The forward process is modified as $q(\mathbf{x}_\tau^{[T]} | \mathbf{x}_{\tau-1}^{[T]}, \mathbf{x}_c^{[T_c]}) := \mathcal{N}(\mathbf{x}_\tau^{[T]}; \mathbf{x}_r + \sqrt{1-\beta_\tau}(\mathbf{x}_{\tau-1}^{[T]} - \mathbf{x}_r), \beta_\tau \mathbf{I})$, which ensures $q(\mathbf{x}_\mathcal{T}^{[T]} | \mathbf{x}_c^{[T_c]})$ matches the equivariant prior $\mathbf{x}_r$ (proof in App. A.2). The reverse transition kernel is given by $p_{\tau-1}(\mathbf{x}_{\tau-1}^{[T]} | \mathbf{x}_\tau^{[T]}, \mathbf{x}_c^{[T_c]}) = \mathcal{N}(\boldsymbol{\mu}_{\boldsymbol{\theta}}(\mathbf{x}_\tau^{[T]}, \tau, \mathbf{x}_c^{[T_c]}), \sigma_\tau^2 \mathbf{I})$. Similar to the unconditional case, we also adopt the noise prediction objective by rewriting $\boldsymbol{\mu}_{\boldsymbol{\theta}}(\mathbf{x}_\tau^{[T]}, \mathbf{x}_c^{[T_c]}, \tau) = \mathbf{x}_r^{[T]} + \frac{1}{\sqrt{\alpha_\tau}}\left(\mathbf{x}_\tau^{[T]} - \mathbf{x}_r^{[T]} - \frac{\beta_\tau}{\sqrt{1-\bar{\alpha}_\tau}}\boldsymbol{\epsilon}_{\boldsymbol{\theta}}(\mathbf{x}_\tau^{[T]}, \mathbf{x}_c^{[T_c]}, \tau)\right)$. The denoising network $\boldsymbol{\epsilon}_{\boldsymbol{\theta}}$ is implemented as an EGTN but with its output subtracted by the input for translation invariance, hence the translation equivariance of $\boldsymbol{\mu}_{\boldsymbol{\theta}}$.

**Training and inference.** Optimizing the VLB of our diffusion yields the following objective:

$$\mathcal{L}_{\text{cond}} := \mathbb{E}_{\mathbf{x}_0^{[T]}, \mathbf{x}_c^{[T_c]}, \boldsymbol{\epsilon} \sim \mathcal{N}(\mathbf{0}, \mathbf{I}), \tau \sim \text{Unif}(1, \mathcal{T})} \left[ \|\boldsymbol{\epsilon} - \boldsymbol{\epsilon}_{\boldsymbol{\theta}}(\mathbf{x}_\tau^{[T]}, \mathbf{x}_c^{[T_c]}, \tau)\|^2 \right], \tag{12}$$

after simplification (proof in App. A.2). The training and inference procedures are in Alg. 3 and 4.

## 5  Experiments

We evaluate GeoTDM on N-body physical simulation, molecular dynamics, and pedestrian trajectory forecasting, in both conditional (§ 5.1) and unconditional generation (§ 5.2) scenarios. We ablate our core design choices and demonstrate additional use cases in § 5.3.

Table 1: Conditional generation on N-body. Results averaged over 5 runs, std in App. C.4.

|  | Particle | | Spring | | Gravity | |
|---|---|---|---|---|---|---|
|  | ADE | FDE | ADE | FDE | ADE | FDE |
| RF [27] | 0.479 | 1.050 | 0.0145 | 0.0389 | 0.791 | 1.630 |
| TFN [56] | 0.330 | 0.754 | 0.1013 | 0.2364 | 0.327 | 0.761 |
| SE(3)-Tr [9] | 0.395 | 0.936 | 0.0865 | 0.2043 | 0.338 | 0.830 |
| EGNN [43] | 0.186 | 0.426 | 0.0101 | 0.0231 | 0.310 | 0.709 |
| EqMotion [62] | 0.141 | 0.310 | 0.0134 | 0.0358 | 0.302 | 0.671 |
| SVAE [67] | 0.378 | 0.732 | 0.0120 | 0.0209 | 0.582 | 1.101 |
| GeoTDM | **0.110** | **0.258** | **0.0030** | **0.0079** | **0.256** | **0.613** |

Table 2: Pedestrian trajectory forecasting on ETH-UCY. Best in **bold** and second best underlined.

|  | ETH | Hotel | Univ | Zara1 | Zara2 | Average |
|---|---|---|---|---|---|---|
| Linear | 1.07/2.28 | 0.31/0.61 | 0.52/1.16 | 0.42/0.95 | 0.32/0.72 | 0.53/1.14 |
| SGAN [14] | 0.64/1.09 | 0.46/0.98 | 0.56/1.18 | 0.33/0.67 | 0.31/0.64 | 0.46/0.91 |
| SoPhie [39] | 0.70/1.43 | 0.76/1.67 | 0.54/1.24 | 0.30/0.63 | 0.38/0.78 | 0.54/1.15 |
| PECNet [32] | 0.54/0.87 | 0.18/0.24 | 0.35/0.60 | 0.22/0.39 | 0.17/0.30 | 0.29/0.48 |
| Traj++ [40] | 0.54/0.94 | 0.16/0.28 | 0.28/0.55 | 0.21/0.42 | 0.16/0.32 | 0.27/0.50 |
| BiTraP [68] | 0.56/0.98 | 0.17/0.28 | 0.25/0.47 | 0.23/0.45 | 0.16/0.33 | 0.27/0.50 |
| MID [12] | 0.50/0.76 | 0.16/0.24 | 0.28/0.49 | 0.25/0.41 | 0.19/0.35 | 0.27/0.45 |
| SVAE [67] | 0.47/0.76 | 0.14/0.22 | 0.25/0.47 | **0.20/0.37** | **0.14/0.28** | 0.24/0.42 |
| GeoTDM | **0.46/0.64** | **0.13/0.21** | **0.24/0.45** | 0.21/0.39 | 0.16/0.30 | **0.23/0.40** |

Table 3: Conditional trajectory generation on MD17. Results averaged over 5 runs (std in App. C.4).

|  | Aspirin | | Benzene | | Ethanol | | Malonaldehyde | | Naphthalene | | Salicylic | | Toluene | | Uracil | |
|---|---|---|---|---|---|---|---|---|---|---|---|---|---|---|---|---|
|  | ADE | FDE | ADE | FDE | ADE | FDE | ADE | FDE | ADE | FDE | ADE | FDE | ADE | FDE | ADE | FDE |
| RF [27] | 0.303 | 0.442 | 0.120 | 0.194 | 0.374 | 0.515 | 0.297 | 0.454 | 0.168 | 0.185 | 0.261 | 0.343 | 0.199 | 0.249 | 0.239 | 0.272 |
| TFN [56] | 0.133 | 0.268 | 0.024 | 0.049 | 0.201 | 0.414 | 0.184 | 0.386 | 0.072 | 0.098 | 0.115 | 0.223 | 0.090 | 0.150 | 0.090 | 0.159 |
| SE(3)-Tr. [9] | 0.294 | 0.556 | 0.027 | 0.056 | 0.188 | 0.359 | 0.214 | 0.456 | 0.069 | 0.103 | 0.189 | 0.312 | 0.108 | 0.184 | 0.107 | 0.196 |
| EGNN [43] | 0.267 | 0.564 | 0.024 | 0.042 | 0.268 | 0.401 | 0.393 | 0.958 | 0.095 | 0.133 | 0.159 | 0.348 | 0.207 | 0.294 | 0.154 | 0.282 |
| EqMotion [62] | 0.185 | 0.246 | 0.029 | 0.043 | 0.152 | 0.247 | 0.155 | 0.249 | 0.073 | 0.092 | 0.110 | 0.151 | 0.097 | 0.129 | 0.088 | 0.116 |
| SVAE [67] | 0.301 | 0.428 | 0.114 | 0.133 | 0.387 | 0.505 | 0.287 | 0.430 | 0.124 | 0.135 | 0.122 | 0.142 | 0.145 | 0.171 | 0.145 | 0.156 |
| GeoTDM | **0.107** | **0.193** | **0.023** | **0.039** | **0.115** | **0.209** | **0.107** | **0.176** | **0.064** | **0.087** | **0.083** | **0.120** | **0.083** | **0.121** | **0.074** | **0.099** |

## 5.1 Conditional Case

### 5.1.1 N-body

**Experimental setup.** We adopt three scenarios in the collection of N-body simulation datasets, including **1.** Charged Particles [26, 43], where $N = 5$ particles with charges randomly chosen between $+1/-1$ are moving under Coulomb force; **2.** Spring Dynamics [26], where $N = 5$ particles with random mass are connected by springs with a probability of 0.5 between each pairs, and force on the spring follows Hooke's law; **3.** Gravity System [2], where $N = 10$ particles with random mass and initial velocity moves driven by gravitational force. For all three datasets, we use 3000 trajectories for training, 2000 for validation, and 2000 for testing. For each trajectory, we use 10 frames as the condition and predict the trajectory for the next 20 frames.

**Baselines.** We involve baselines from three families. Frame-to-frame prediction models: Radial Field [27], Tensor Field Network [56], SE(3)-Transformer [9], and EGNN [43]; Deterministic trajectory model: EqMotion [62]; Probabilistic trajectory model: SVAE [67]. Details in App. B.3.

**Metrics.** We employ Average Discrepancy Error (ADE) and Final Discrepancy Error (FDE), which are widely adopted for trajectory forecasting [67, 62], given by $\text{ADE}(\mathbf{x}^{[T]}, \mathbf{y}^{[T]}) = \frac{1}{TN} \sum_{t=0}^{T-1} \sum_{i=0}^{N-1} \|\mathbf{x}_i^{(t)} - \mathbf{y}_i^{(t)}\|_2$, and $\text{FDE}(\mathbf{x}^{[T]}, \mathbf{y}^{[T]}) = \frac{1}{N} \sum_{i=0}^{N-1} \|\mathbf{x}_i^{(T-1)} - \mathbf{y}_i^{(T-1)}\|_2$. For probabilistic models, we report average ADE and FDE derived from $K = 5$ samples.

**Implementation.** The input data are processed as geometric graphs. For example, on Charged Particles, the node feature is the charge, and the graph is specified as fully connected without self-loops. We use 6 layers in EGTN with hidden dimension of 128. We use $\mathcal{T} = 1000$ and the linear noise schedule [18]. More details in App. B.2.

**Results.** We present the results in Table 1, with the following observations. **1.** Trajectory models generally yield lower error than frame to frame prediction models since they mitigate errors accumulated in iterative roll-out. **2.** The equivariant methods, *e.g.*, EGNN, EqMotion, and our GeoTDM significantly improves over the non-equivariant model SVAE, demonstrating the importance of injecting physical symmetry into the modeling of geometric trajectories. **3.** By directly modeling the distribution of geometric trajectories with equivariance, GeoTDM achieves the lowest ADE and FDE on all three tasks, showcasing the superiority of the proposed approach.

### 5.1.2 Molecular Dynamics

**Experimental setup.** We employ the MD17 [5] dataset, which contains the DFT-simulated molecular dynamics (MD) trajectories of 8 small molecules, with the number of atoms for each molecule

Table 4: MD Trajectory generation results on MD17. Marg, Class, and Pred refer to Marginal score, Classification score, and Prediction score respectively. GeoTDM performs the best on all 8 molecules.

| | Aspirin | | | Benzene | | | Ethanol | | | Malonaldehyde | | |
|---|---|---|---|---|---|---|---|---|---|---|---|---|
| | Marg ↓ | Class ↑ | Pred ↓ | Marg ↓ | Class ↑ | Pred ↓ | Marg ↓ | Class ↑ | Pred ↓ | Marg ↓ | Class ↑ | Pred ↓ |
| SVAE [67] | 3.628 | $6.80\times10^{-5}$ | 0.0949 | 4.755 | $2.81\times10^{-6}$ | 0.0181 | 2.735 | $2.39\times10^{-5}$ | 0.0929 | 2.808 | $5.57\times10^{-3}$ | 0.0346 |
| EGVAE [43] | 2.650 | $1.31\times10^{-4}$ | 0.0386 | 3.677 | $1.50\times10^{-4}$ | 0.0104 | 2.617 | $5.86\times10^{-6}$ | 0.1131 | 2.767 | $1.73\times10^{-6}$ | 0.0664 |
| GeoTDM | **0.726** | $\mathbf{3.48\times10^{-2}}$ | **0.0212** | **0.597** | $\mathbf{1.62\times10^{-1}}$ | **0.0019** | **0.314** | $\mathbf{4.63\times10^{-1}}$ | **0.0235** | **0.403** | $\mathbf{3.35\times10^{-1}}$ | **0.0146** |

| | Naphthalene | | | Salicylic | | | Toluene | | | Uracil | | |
|---|---|---|---|---|---|---|---|---|---|---|---|---|
| | Marg ↓ | Class ↑ | Pred ↓ | Marg ↓ | Class ↑ | Pred ↓ | Marg ↓ | Class ↑ | Pred ↓ | Marg ↓ | Class ↑ | Pred ↓ |
| SVAE [67] | 3.150 | $2.50\times10^{-2}$ | 0.2123 | 2.941 | $3.54\times10^{-6}$ | 0.1312 | 3.083 | $8.29\times10^{-5}$ | 0.2580 | 2.736 | $3.73\times10^{-5}$ | 0.604 |
| EGVAE [43] | 3.007 | $3.17\times10^{-4}$ | 0.0136 | 3.314 | $3.76\times10^{-6}$ | 0.0221 | 2.054 | $2.77\times10^{-5}$ | 0.0457 | 3.570 | $2.02\times10^{-5}$ | 0.0212 |
| GeoTDM | **0.770** | $\mathbf{1.17\times10^{-1}}$ | **0.0093** | **0.559** | $\mathbf{1.82\times10^{-1}}$ | **0.0135** | **0.539** | $\mathbf{1.12\times10^{-1}}$ | **0.0118** | **0.954** | $\mathbf{2.02\times10^{-1}}$ | **0.0116** |

ranging from 9 (Ethanol and Malonaldehyde) to 21 (Aspirin). For each molecule, we construct a training set of 5000 trajectories, and 1000/1000 for validation and testing, uniformly sampled along the time dimension. Different from [62], we explicitly involve the hydrogen atoms which contribute most to the vibrations of the trajectory, leading to a more challenging task. The node feature is the one-hot encodings of atomic number [44] and edges are connected between atoms within three hops measured in atomic bonds [48]. We adopt the same set of baselines as the N-body experiments.

**Results.** As depicted in Table 3, GeoTDM achieves the best performance on all eight molecule MD trajectories, outperforming previos state-of-the-art approach EqMotion. In particular, GeoTDM obtains an improvement of 23.1%/15.3% on average in terms of ADE/FDE, compared with the previous state-of-the-art approach EqMotion, thanks to the probabilistic modeling which is advantageous in capturing the stochasticity of MD simulations.

### 5.1.3 Pedestrian Trajectory Forecasting

**Experimental setup.** We apply our model to ETH-UCY [35, 28] dataset, a challenging and large-scale benchmark for pedestrian trajectory forecasting. There are five scenes in total: ETH, Hotel, Univ, Zara1, and Zara2. Following standard setup [14, 67], we use 8 frames (3.2 seconds) as input to predict the next 12 frames (4.8 seconds). The pedestrians are viewed as nodes and their 2D coordinates are extracted from the scenes. Edges are connected for nodes within a preset distance measured from the final frame in the given trajectory. The metrics are minADE/minFDE computed from 20 samples. For baselines, we compare with existing generative models that have been specifically designed for pedestrian trajectory prediction, including GANs: SGAN [14], SoPhie [39]; VAEs: PECNet [32], Traj++ [40], BiTraP [68], SVAE [67]; and diffusion: MID [12]. Baseline results are taken from [67].

**Results.** From Table 2, we observe that our GeoTDM obtains the best predictions on 3 out of the 5 scenarios while achieving the lowest average ADE and FDE. It is remarkable since compared with these baselines specifically tailored for the task of pedestrian trajectory forecasting, GeoTDM does not involve special data preprocessing of the trajectories through rotations or translations, does not involve extra auxiliary losses to optimize during training, and does not require task-specific backbones, demonstrating its general effectiveness across different geometric domains.

### 5.2 Unconditional Generation

**Experimental setup.** For generation we reuse the Charged Particle dataset and the MD17 dataset. We follow the same setup as the conditional case, except that we generate trajectories with length 20 from scratch. We compare with SGAN [14], SVAE [67] (slightly modified to enable generation from scratch), and a VAE-modified version of EGNN [43], dubbed EGVAE (see App. B). The results of SGAN on MD17 is omitted due to mode collapse during training.

**Metrics.** We adopt three metrics adapted from time series generation to quantify the generation quality of the geometric trajectories: *Marginal* scores [34] measure the distance between the empirical probability density functions of the generated samples and the ground truths; *Classification* scores [25] are computed as the cross-entropy loss given by a trajectory classification model, trained on the task of distinguish whether the trajectory is generated or real; *Prediction* scores [72] are the MSEs of a train-on-synthetic-test-on-real trajectory prediction model (a 1-layer EqMotion) that takes as input the first half of the trajectories to predict the other half.

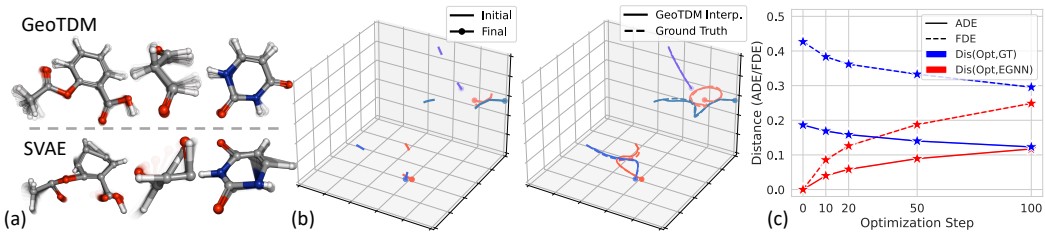

Figure 2: (a) Unconditional generation samples on MD17. GeoTDM generates MD trajectories with much higher quality (see more in App. D). (b) Interpolation. *Left*: the given initial and final 5 frames. *Right*: GeoTDM interpolation and GT. (c) Optimization by GeoTDM on predictions of EGNN. Dis(Opt, GT)/Dis(Opt, EGNN) is the distance between optimized trajectories and GT/EGNN.

**Results.** Quantitative results are displayed in Table 5 and 4 for N-body and MD17. Notably, GeoTDM delivers samples with much higher quality than the baselines. On Charged Particles, GeoTDM achieves a classification score of 0.556, indicating its generated samples are generally indistinguishable with the ground truths. We observe similar patterns on MD17, where GeoTDM obtains remarkably lower marginal scores, higher classification scores, and lower prediction scores, showcasing its strong capability to model complex distributions of geometric trajectories on various geometric data. Visualizations are in Fig. 5 and more in App. D.

Table 5: Unconditional generation results on N-body Charged Particle.

|  | Marg $\downarrow$ | Class $\uparrow$ | Pred $\downarrow$ |
|---|---|---|---|
| SGAN [14] | 0.1448 | $3.98\times10^{-7}$ | 0.172 |
| SVAE [67] | 0.0668 | $1.38\times10^{-6}$ | 0.282 |
| EGVAE [43] | 0.1141 | $4.22\times10^{-2}$ | 0.0467 |
| GeoTDM | **0.0055** | $\mathbf{5.56\times10^{-1}}$ | **0.00978** |

## 5.3 Ablation Studies and Additional Use Cases

**Ablations on diffusion prior.** We investigate different priors, including non-equivariant $\mathcal{N}(\mathbf{0}, \mathbf{I})$ (*i.e.*, DDPM [18]), equivariant but fixed CoM prior $\mathcal{N}(\text{CoM}(\mathbf{x}_c^{(T_c-1)}), \mathbf{I})$ and point-wise equivariant prior $\mathcal{N}(\mathbf{x}_c^{(T_c-1)}, \mathbf{I})$. In Table 6 we see that non-equivariant prior leads to significantly worse performance. The CoM prior, though equivariant, is still inferior due to extra overhead in denoising the nodes initialized around the CoM to the original geometry. GeoTDM yields the lowest error due to the flexible learnable prior.

Table 6: Ablation studies. The numbers refer to ADE/FDE.

|  | Charge | Aspirin |
|---|---|---|
| GeoTDM $\mathcal{N}(\mathbf{x}_r^{[T]}, \mathbf{I})$ | **0.110/0.258** | **0.107/0.193** |
| Fixed $\mathcal{N}(\mathbf{0}, \mathbf{I})$ | 0.220/0.485 | 0.235/0.393 |
| $\mathcal{N}(\text{CoM}(\mathbf{x}_c^{(T_c-1)}), \mathbf{I})$ | 0.135/0.298 | 0.119/0.212 |
| $\mathcal{N}(\mathbf{x}_c^{(T_c-1)}, \mathbf{I})$ | 0.123/0.282 | 0.110/0.204 |
| w/o Equivariance | 0.251/0.542 | 0.252/0.440 |
| w/o Attention | 0.133/0.312 | 0.114/0.208 |
| w/o Shift invariance | 0.139/0.330 | 0.112/0.212 |

**Ablations on EGTN.** We further ablate the designs of the denoising model. **1.** Equivariance. We replace all EGCL layers into non-equivariant MPNN [11] layers with same hidden dimension, leading to non-equivariant transition kernels. The performance becomes much worse, verifying the necessity of equivariance. **2.** Attention. We substitute the attentions in temporal layers by equivariant convolutions (see App. B). Compared with this variant, GeoTDM enjoys larger capacity with attention and yields lower prediction error especially on Charged Particle where the particles generally move faster. **3.** Temporal shift invariance. We employ relative temporal embeddings in attention, which enhances the generalization. Notably, the FDE improves from 0.330 to 0.258 on Charged Particle compared with the absolute temporal embedding.

**Temporal interpolation.** GeoTDM is able to perform interpolation as a special case of the conditional case. We demonstrate such capability on Charged Particle. The model is provided with the first 5 and last 5 frames, and the task is to generate the intermediate 20 frames as interpolation. GeoTDM reports an ADE of 0.055 on the test set, while a linear interpolation baseline reports an ADE of 0.171. From the qualitative visualizations in Fig. 2, we clearly see that GeoTDM can capture the complex dynamics and yield high-quality non-linear interpolations between the given initial and final frames.

**Optimization.** We further illustrate that GeoTDM can conduct optimization [31, 33] on given trajectories (*e.g.*, those simulated by an EGNN) by simulating $K$ steps through the forward diffusion and then performing the reverse denoising. From Fig. 2 we see the distance between the optimized trajectory and GT gradually decreases as the optimization step grows. This reveals GeoTDM can effectively optimize the given trajectory towards the ground truth distribution.

# 6 Discussion

**Limitations.** Akin to other diffusion models, GeoTDM resorts to multi-step sampling which may require more compute. We present empirical runtime benchmarks and more discussions in App. C.3.

**Conclusion.** We present GeoTDM, a diffusion model built over distribution of geometric trajectories. It is designed to preserve the symmetry of geometric systems, achieved by using EGTN, a novel SE(3)-equivariant geometric trajectory model, as the denoising network. We evaluate GeoTDM on various datasets for unconditional generation, interpolation, extrapolation and optimization, showing that it consistently outperforms the baselines. Future works include streamlining GeoTDM and extending it to more tasks such as protein MD, robot manipulation, and motion synthesis.

## Acknowledgments and Disclosure of Funding

We thank the anonymous reviewers for the helpful feedback on improving the manuscript. This work was supported by ARO (W911NF-21-1-0125), ONR (N00014-23-1-2159), and the CZ Biohub.

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

# Appendix

## Table of Contents

## A  Proofs

### A.1  Unconditional Case

We note that, naïvely, a distribution $p(\mathbf{x}^{[T]})$ can not be translation invariant. In particular, this would imply that $p(\mathbf{x}^{[T]}) = p(\mathbf{x}^{[T]} + \mathbf{r})$ for all $\mathbf{r} \in \mathbb{R}^D$, but this would imply that $p(\mathbf{x}^{[T]}) = 0$ uniformly, a contradiction.

Instead, we derive an equivalent invariance condition by restricting $\mathrm{SE}(D)$ to its maximally compact subgroup. In particular, we note that it is possible to define $\mathrm{SO}(D)$-invariant distributions (as this group is compact), and $\mathrm{SE}(D)/\mathrm{SO}(D) \cong \mathbb{T}$, the translation group. The natural way to quotient out our base space $\mathbb{R}^{T \times N \times D}/\mathbb{T} \cong \mathbb{R}^{(T \times N - 1) \times D}$ is to zero-center our data (along each dimension).

However, for practical purposes, we will refer to our construction as $\mathrm{SE}(D)$-invariant. In particular, since all inputs $\mathbf{x} \in \mathbb{R}^{T \times N \times D}$ are first zero-centered to be projected to $\mathbb{R}^{T \times N \times D}/\mathbb{T}$, the "lifted" unnormalized measure is $\mathrm{SE}(D)$ invariant.

We will define $P$ as our zero-centering operation $P(\mathbf{x}^{[T]}) = \mathbf{x}^{[T]} - \frac{1}{T}\sum_{t=0}^{T-1}\mathrm{CoM}(\mathbf{x}^{(t)})$, with $\mathrm{CoM}(\mathbf{x}^{(t)}) = \frac{1}{N}\sum_{i=1}^{N}\mathbf{x}_i^{(t)}$ and our restricted $(T \times N - 1) \times D$ Gaussian as $\tilde{\mathcal{N}}(\mathbf{y}|\mathbf{x}, \boldsymbol{\Sigma})$, which can be represented in the ambient space as a degenerated Gaussian variable

$$\tilde{\mathcal{N}}(\mathbf{y}|\mathbf{x}, \boldsymbol{\Sigma}) = \frac{1}{(2\pi)^{2/((T \times N - 1) \times D)}\det^*(\boldsymbol{\Sigma_P})^{1/2}}\exp\left(-\frac{1}{2}(\mathbf{y}-\mathbf{x})^\top\boldsymbol{\Sigma_P}^+(\mathbf{y}-\mathbf{x})\right) \quad (13)$$

where $\boldsymbol{\Sigma_P} = \mathbf{P}\boldsymbol{\Sigma}\mathbf{P}^\top$ and $\boldsymbol{\Sigma_P}^+$ is the pseudo-inverse (and $\det^*$ is the determinant restricted to the subspace). Note that $\mathbf{P}$ is symmetric and idempotent. Then specifically when $\boldsymbol{\Sigma} = \mathbf{I}$, we have $\boldsymbol{\Sigma_P} = \mathbf{P}\mathbf{P}^\top$, then $\boldsymbol{\Sigma_P}^+ = \mathbf{P}$ as $(\mathbf{P}\mathbf{P}^\top)\mathbf{P}(\mathbf{P}\mathbf{P}^\top) = \mathbf{P}$, since $\mathbf{P}\mathbf{P} = \mathbf{P}$ and $\mathbf{P} = \mathbf{P}^\top$.

**Base distribution.** We require the base distribution to be SO(3)-invariant. In practice, we let $p_{\mathcal{T}}(\tilde{\mathbf{x}}_{\mathcal{T}}^{[T]}) = \tilde{\mathcal{N}}(\mathbf{0}, \mathbf{I})$ to be the Gaussian distribution in the translation-invariant subspace

**Transition kernel.** For the transition kernel, we specify it as $p_{\theta}(\tilde{\mathbf{x}}_{\tau-1}^{[T]} \mid \tilde{\mathbf{x}}_{\tau}^{[T]}) = \tilde{\mathcal{N}}(\tilde{\boldsymbol{\mu}}_{\theta}(\tilde{\mathbf{x}}_{\tau}^{[T]}, \tau), \sigma_{\tau}^2 \mathbf{I})$. In order to ensure $p_{\theta}(\mathbf{R}\tilde{\mathbf{x}}_{\tau-1}^{[T]} \mid \mathbf{R}\tilde{\mathbf{x}}_{\tau}^{[T]}) = p_{\theta}(\tilde{\mathbf{x}}_{\tau-1}^{[T]} \mid \tilde{\mathbf{x}}_{\tau}^{[T]})$, it suffices to make $\tilde{\boldsymbol{\mu}}_{\theta}(\tilde{\mathbf{x}}_{\tau}^{[T]}, \tau)$ an SO(3)-equivariant function. In this way,

$$p_{\theta}(\mathbf{R}\tilde{\mathbf{x}}_{\tau-1}^{[T]} \mid \mathbf{R}\tilde{\mathbf{x}}_{\tau}^{[T]}) = \tilde{\mathcal{N}}(\mathbf{R}\tilde{\mathbf{x}}_{\tau-1}^{[T]}; \tilde{\boldsymbol{\mu}}_{\theta}(\mathbf{R}\tilde{\mathbf{x}}_{\tau}^{[T]}), \sigma_{\tau}^2 \mathbf{I}), \tag{14}$$

$$= \tilde{\mathcal{N}}(\mathbf{R}\tilde{\mathbf{x}}_{\tau-1}^{[T]} - \tilde{\boldsymbol{\mu}}_{\theta}(\mathbf{R}\tilde{\mathbf{x}}_{\tau}^{[T]}); \mathbf{0}, \sigma_{\tau}^2 \mathbf{I}), \tag{15}$$

$$= \tilde{\mathcal{N}}(\mathbf{R}\tilde{\mathbf{x}}_{\tau-1}^{[T]} - \mathbf{R}\tilde{\boldsymbol{\mu}}_{\theta}(\tilde{\mathbf{x}}_{\tau}^{[T]}); \mathbf{0}, \sigma_{\tau}^2 \mathbf{I}), \tag{16}$$

$$= \tilde{\mathcal{N}}(\mathbf{R}(\tilde{\mathbf{x}}_{\tau-1}^{[T]} - \tilde{\boldsymbol{\mu}}_{\theta}(\tilde{\mathbf{x}}_{\tau}^{[T]})); \mathbf{0}, \sigma_{\tau}^2 \mathbf{I}), \tag{17}$$

$$= \tilde{\mathcal{N}}(\tilde{\mathbf{x}}_{\tau-1}^{[T]} - \tilde{\boldsymbol{\mu}}_{\theta}(\tilde{\mathbf{x}}_{\tau}^{[T]}); \mathbf{0}, \sigma_{\tau}^2 \mathbf{I}), \tag{18}$$

$$= \tilde{\mathcal{N}}(\tilde{\mathbf{x}}_{\tau-1}^{[T]}; \tilde{\boldsymbol{\mu}}_{\theta}(\tilde{\mathbf{x}}_{\tau}^{[T]}), \sigma_{\tau}^2 \mathbf{I}), \tag{19}$$

$$= p_{\theta}(\tilde{\mathbf{x}}_{\tau-1}^{[T]} \mid \tilde{\mathbf{x}}_{\tau}^{[T]}), \tag{20}$$

which permits the SO(3)-equivariance of the transition kernel. In our implementation, we further re-parameterize $\tilde{\boldsymbol{\mu}}_{\theta}(\tilde{\mathbf{x}}_{\tau}^{[T]}, \tau)$ as,

$$\tilde{\boldsymbol{\mu}}_{\theta}(\tilde{\mathbf{x}}_{\tau}^{[T]}, \tau) = \frac{1}{\sqrt{\alpha_{\tau}}} \left( \tilde{\mathbf{x}}_{\tau}^{[T]} - \frac{\beta_{\tau}}{\sqrt{1 - \bar{\alpha}_{\tau}}} \tilde{\boldsymbol{\epsilon}}_{\theta}(\tilde{\mathbf{x}}_{\tau}^{[T]}, \tau) \right), \tag{21}$$

where we instead ensure $\tilde{\boldsymbol{\epsilon}}_{\theta}(\tilde{\mathbf{x}}_{\tau}^{[T]}, \tau)$ to be SO(3)-equivariant and its output should lie in the subspace $\mathcal{X}_{\mathbf{P}}$.

We now prove the following proposition, which states that if the base distribution is SO(3)-invariant and the transition kernel is SO(3)-equivariant, then the marginal at any diffusion time step is also SO(3)-invariant.

**Proposition A.1.** *If the prior $p_{\mathcal{T}}(\tilde{\mathbf{x}}_{\mathcal{T}}^{[T]})$ is SO(3)-invariant, the transition kernels $p_{\tau-1}(\tilde{\mathbf{x}}_{\tau-1}^{[T]} \mid \tilde{\mathbf{x}}_{\tau}^{[T]}), \forall \tau \in \{1, \cdots, \mathcal{T}\}$ are SO(3)-equivariant, then the marginal $p_{\tau}(\tilde{\mathbf{x}}_{\tau}^{[T]})$ at any time step $\tau \in \{0, \cdots, \mathcal{T}\}$ is also SO(3)-invariant.*

*Proof.* The proof is given by induction.

**Induction base.** When $\tau = \mathcal{T}$, we have the marginal being the prior $p_{\mathcal{T}}(\tilde{\mathbf{x}}_{\mathcal{T}}^{[T]})$, which is SO(3)-invariant.

**Induction step.** Suppose the marginal at diffusion time step $\tau$ is SO(3)-invariant, *i.e.*, $p_{\tau}(\tilde{\mathbf{x}}_{\tau}^{[T]}) = p_{\tau}(\mathbf{R}\tilde{\mathbf{x}}_{\tau}^{[T]})$, then we have the following derivation for the marginal at time step $\tau - 1$:

$$p_{\tau-1}(\mathbf{R}\tilde{\mathbf{x}}_{\tau-1}^{[T]}) = \int p_{\tau-1}(\mathbf{R}\tilde{\mathbf{x}}_{\tau-1}^{[T]} \mid \tilde{\mathbf{x}}_{\tau}^{[T]}) p_{\tau}(\tilde{\mathbf{x}}_{\tau}^{[T]}) d\tilde{\mathbf{x}}_{\tau}^{[T]}, \tag{22}$$

$$= \int p_{\tau-1}(\mathbf{R}\tilde{\mathbf{x}}_{\tau-1}^{[T]} \mid \mathbf{R}\mathbf{R}^{-1}\tilde{\mathbf{x}}_{\tau}^{[T]}) p_{\tau}(\mathbf{R}\mathbf{R}^{-1}\tilde{\mathbf{x}}_{\tau}^{[T]}) d\tilde{\mathbf{x}}_{\tau}^{[T]}, \tag{23}$$

$$= \int p_{\tau-1}(\tilde{\mathbf{x}}_{\tau-1}^{[T]} \mid \mathbf{R}^{-1}\tilde{\mathbf{x}}_{\tau}^{[T]}) p_{\tau}(\mathbf{R}^{-1}\tilde{\mathbf{x}}_{\tau}^{[T]}) d\tilde{\mathbf{x}}_{\tau}^{[T]}, \tag{24}$$

$$= \int p_{\tau-1}(\tilde{\mathbf{x}}_{\tau-1}^{[T]} \mid \tilde{\mathbf{y}}_{\tau}^{[T]}) p_{\tau}(\tilde{\mathbf{y}}_{\tau}^{[T]}) \det(\mathbf{R}) d\tilde{\mathbf{y}}_{\tau}^{[T]}, \tag{25}$$

$$= p_{\tau-1}(\tilde{\mathbf{x}}_{\tau-1}^{[T]}). \tag{26}$$

$\square$

Notably, for the final step at $\tau = 0$, the marginal $p_0(\mathbf{R}\tilde{\mathbf{x}}_0^{[T]})$ is also SO(3)-invariant, indicating the final sample from the entire geometric trajectory diffusion process resides in an SO(3)-invariant distribution, hence the physical symmetry being well preserved.

**Algorithm 1** Training Procedure of GeoTDM-uncond

1: **repeat**
2:  Sample $\tilde{\boldsymbol{\epsilon}}^{[T]} \sim \tilde{\mathcal{N}}(\mathbf{0}, \mathbf{I})^{[T]}, \tau \in \text{Unif}(\{1, \cdots, \mathcal{T}\})$,
   $\quad\quad \tilde{\mathbf{x}}^{[T]} \sim \tilde{\mathcal{D}}_{\text{data}}$
3:  $\tilde{\mathbf{x}}_\tau^{[T]} \leftarrow \sqrt{\bar{\alpha}_\tau}\tilde{\mathbf{x}}^{[T]} + \sqrt{1 - \bar{\alpha}_\tau}\tilde{\boldsymbol{\epsilon}}^{[T]}$
4:  Take gradient descent step on
   $\quad\quad \nabla_{\boldsymbol{\theta}}\|\tilde{\boldsymbol{\epsilon}}^{[T]} - \tilde{\boldsymbol{\epsilon}}_{\boldsymbol{\theta}}(\tilde{\mathbf{x}}_\tau^{[T]}, \tau)\|_2^2$
5: **until** converged

---

**Algorithm 2** Sampling Procedure of GeoTDM-uncond

1: Sample $\tilde{\mathbf{x}}_{\mathcal{T}}^{[T]} \sim \tilde{\mathcal{N}}(\mathbf{0}, \mathbf{I})^{[T]}$
2: **for** $\tau \leftarrow \mathcal{T}, \cdots, 1$ **do**
3:  Sample $\tilde{\mathbf{z}}_\tau^{[T]} \sim \tilde{\mathcal{N}}(\mathbf{0}, \mathbf{I})^{[T]}$ if $\tau > 1$ else $\tilde{\mathbf{z}}_\tau^{[T]} = \mathbf{0}$
4:  $\tilde{\mathbf{x}}_{\tau-1}^{[T]} \leftarrow \frac{1}{\sqrt{\alpha_\tau}}\left(\tilde{\mathbf{x}}_\tau^{[T]} - \frac{1-\alpha_\tau}{\sqrt{1-\bar{\alpha}_\tau}}\tilde{\boldsymbol{\epsilon}}_{\boldsymbol{\theta}}(\tilde{\mathbf{x}}_\tau^{[T]}, \tau)\right) + \sigma_\tau\tilde{\mathbf{z}}_\tau^{[T]}$
5: **end for**
6: **return** $\mathbf{x}_0^{[T]}$

---

**Algorithm 3** Training Procedure of GeoTDM-cond

1: **repeat**
2:  $\mathbf{x}_r^{[T]} \leftarrow \text{EquiPrior}_{\boldsymbol{\eta}, \boldsymbol{\gamma}}(\mathbf{x}_c^{[T_c]})$     {Eq. 9}
3:  Sample $\boldsymbol{\epsilon}^{[T]} \sim \mathcal{N}(\mathbf{0}, \mathbf{I}), \tau \in \text{Unif}(\{1, \cdots, \mathcal{T}\})$,
   $\quad\quad (\mathbf{x}^{[T]}, \mathbf{x}_c^{[T_c]},) \sim p_{\text{data}}$
4:  $\mathbf{x}_\tau^{[T]} \leftarrow \sqrt{\bar{\alpha}_\tau}(\mathbf{x}^{[T]} - \mathbf{x}_r^{[T]}) + \mathbf{x}_r^{[T]} + \sqrt{1 - \bar{\alpha}_\tau}\boldsymbol{\epsilon}^{[T]}$
5:  Take gradient descent step on
   $\quad\quad \nabla_{\boldsymbol{\theta}, \boldsymbol{\eta}, \boldsymbol{\gamma}}\|\boldsymbol{\epsilon}^{[T]} - \boldsymbol{\epsilon}_{\boldsymbol{\theta}}(\mathbf{x}_\tau^{[T]}, \mathbf{x}_c^{[T_c]}, \tau)\|^2$
6: **until** converged

---

**Algorithm 4** Sampling Procedure of GeoTDM-cond

1: $\mathbf{x}_r^{[T]} \leftarrow \text{EquiPrior}_{\boldsymbol{\eta}, \boldsymbol{\gamma}}(\mathbf{x}_c^{[T_c]})$     {Eq. 9}
2: Sample $\mathbf{x}_{\mathcal{T}}^{[T]} \sim \mathcal{N}(\mathbf{x}_r^{[T]}, \mathbf{I}), \mathbf{x}_c^{[T_c]} \sim \mathcal{D}_{\text{data}}$
3: **for** $\tau \leftarrow \mathcal{T}, \cdots, 1$ **do**
4:  Sample $\mathbf{z}_\tau^{[T]} \sim \mathcal{N}(\mathbf{0}, \mathbf{I})^{[T]}$ if $\tau > 1$ else $\mathbf{z}_\tau^{[T]} = \mathbf{0}$
5:  $\mathbf{x}_{\tau-1}^{[T]} \leftarrow \frac{1}{\sqrt{\alpha_\tau}}\left(\mathbf{x}_\tau^{[T]} - \mathbf{x}_r^{[T]} - \frac{1-\alpha_\tau}{\sqrt{1-\bar{\alpha}_\tau}}\boldsymbol{\epsilon}_{\boldsymbol{\theta}}(\mathbf{x}_\tau^{[T]}, \mathbf{x}_c^{[T_c]}, \tau)\right) + \mathbf{x}_r^{[T]} + \sigma_\tau\mathbf{z}_\tau^{[T]}$
6: **end for**
7: **return** $\mathbf{x}_0^{[T]}$

---

## A.2  Conditional Case

In the conditional case, we target on modeling the conditional distribution $p(\mathbf{x}^{[T]} \mid \mathbf{x}_c^{[T_c]})$. The desired constraint is the following equivariance condition: $p(\mathbf{x}^{[T]} \mid \mathbf{x}_c^{[T_c]}) = p(g \cdot \mathbf{x}^{[T]} \mid g \cdot \mathbf{x}_c^{[T_c]})$, for all $g \in \text{SE}(3)$.

**Construction of the equivariant prior.** The prior is constructed through Eq. 9. Here we formally show that this guarantees SE(3)-equivariance of the prior. For convenience we repeat Eq. 9 below.

$$\mathbf{x}_r^{(t)} = \sum_{s \in [T_c]} \mathbf{w}^{(t,s)}\hat{\mathbf{x}}_c^{(s)}, \quad \text{s.t.} \sum_{s \in [T_c]} \mathbf{w}^{(t,s)} = \mathbf{1}, \tag{27}$$

Then, we have

$$\mathbf{x}_r'^{(t)} = \sum_{s \in [T_c]} \mathbf{w}'^{(t,s)} \hat{\mathbf{x}}_c'^{(s)}, \tag{28}$$

$$= \sum_{s \in [T_c]} \mathbf{w}^{(t,s)} (\mathbf{R}\hat{\mathbf{x}}_c^{(s)} + \mathbf{r}), \tag{29}$$

$$= \sum_{s \in [T_c]} \mathbf{w}^{(t,s)} (\mathbf{R}\hat{\mathbf{x}}_c^{(s)}) + \sum_{s \in [T_c]} \mathbf{w}^{(t,s)} \mathbf{r}, \tag{30}$$

$$= \mathbf{R} \sum_{s \in [T_c]} \mathbf{w}^{(t,s)} \hat{\mathbf{x}}_c^{(s)} + \mathbf{r}, \tag{31}$$

$$= \mathbf{R}\mathbf{x}_r^{(t)} + \mathbf{r}, \tag{32}$$

$\forall$ rotation matrix $\mathbf{R}$ and $\mathbf{r} \in \mathbb{R}^3$, which completes the proof.

**Base distribution.** We propose to leverage the following base distribution.

$$p_{\mathcal{T}}(\mathbf{x}_{\mathcal{T}}^{[T]} \mid \mathbf{x}_c^{[T_c]}) = \mathcal{N}(\mathbf{x}_{\mathcal{T}}^{[T]}; \mathbf{x}_r^{[T]}, \mathbf{I}), \tag{33}$$

where $\mathbf{x}_r^{[T]} = \mathrm{EquiPrior}(\mathbf{x}_c^{[T_c]})$ is SE(3)-equivariant with respect to the condition $\mathbf{x}_c^{[T_c]}$. With such choice, the base distribution above is SE(3)-equivariant, since

$$p_{\mathcal{T}}(\mathbf{R}\mathbf{x}_{\mathcal{T}}^{[T]} + \mathbf{r} \mid \mathbf{R}\mathbf{x}_c^{[T]} + \mathbf{r}) = \mathcal{N}(\mathbf{R}\mathbf{x}_{\mathcal{T}}^{[T]} + \mathbf{r}; \mathbf{R}\mathbf{x}_r^{[T]} + \mathbf{r}, \mathbf{I}), \tag{34}$$

$$= \mathcal{N}(\mathbf{R}\mathbf{x}_{\mathcal{T}}^{[T]}; \mathbf{R}\mathbf{x}_r^{[T]}, \mathbf{I}), \tag{35}$$

$$= \mathcal{N}(\mathbf{x}_{\mathcal{T}}^{[T]}; \mathbf{x}_r^{[T_c]}, \mathbf{I}), \tag{36}$$

where the last equation is due to $\det(\mathbf{R}^\top \mathbf{R}) = \mathbf{I}$, $\|\mathbf{x}_{\mathcal{T}}^{[T]} - \mathbf{x}_r^{[T]}\|^2 = \|\mathbf{R}\mathbf{x}_{\mathcal{T}}^{[T]} - \mathbf{R}\mathbf{x}_r^{[T]}\|^2$, which also gives the proof for Theorem 4.4 by a mild substitution of the notations.

**Transition kernel.** The transition kernel is given by

$$p_\theta(\mathbf{x}_{\tau-1}^{[T]} \mid \mathbf{x}_\tau^{[T]}, \mathbf{x}_c^{[T_c]}) = \mathcal{N}(\mathbf{x}_{\tau-1}^{[T]}; \boldsymbol{\mu}_\theta(\mathbf{x}_\tau^{[T]}, \mathbf{x}_c^{[T_c]}, \tau), \sigma_\tau^2 \mathbf{I}), \tag{37}$$

where $\boldsymbol{\mu}_\theta(\mathbf{x}_\tau^{[T]}, \mathbf{x}_c^{[T_c]}, \tau)$ parameterized to be SE(3)-equivariant with respect to its input $\mathbf{x}_\tau^{[T]}, \mathbf{x}_c^{[T_c]}$. In practice, we re-parameterize it as,

$$\boldsymbol{\mu}_\theta(\mathbf{x}_\tau^{[T]}, \mathbf{x}_c^{[T_c]}, \tau) = \mathbf{x}_r^{[T]} + \frac{1}{\sqrt{\alpha_\tau}} \left( \mathbf{x}_\tau^{[T]} - \mathbf{x}_r^{[T]} - \frac{\beta_\tau}{\sqrt{1 - \bar{\alpha}_\tau}} \boldsymbol{\epsilon}_\theta(\mathbf{x}_\tau^{[T]}, \mathbf{x}_c^{[T_c]}, \tau) \right), \tag{38}$$

where $\boldsymbol{\epsilon}_\theta(\mathbf{x}_\tau^{[T]}, \mathbf{x}_c^{[T_c]}, \tau)$ is an SO(3)-equivariant but translation-invariant function. It is then easy to see that $\boldsymbol{\mu}_\theta(\mathbf{x}_\tau^{[T]}, \mathbf{x}_c^{[T_c]}, \tau)$ meets the SE(3)-equivariance as desired.

**Proposition A.2.** *With such parameterization, optimizing the variational lower bound is equivalent to optimizing the following objective, up to certain re-weighting:*

$$\mathcal{L} = \|\boldsymbol{\epsilon}_\theta(\mathbf{x}_\tau^{[T]}, \mathbf{x}_c^{[T_c]}, \tau) - \boldsymbol{\epsilon}\|_2^2. \tag{39}$$

*Proof.* We define $q(\mathbf{x}_\tau^{[T]} | \mathbf{x}_{\tau-1}^{[T]}) := \mathcal{N}(\mathbf{x}_\tau^{[T]}; \mathbf{x}_r + \sqrt{1 - \beta_\tau}(\mathbf{x}_{\tau-1}^{[T]} - \mathbf{x}_r), \beta_\tau \mathbf{I})$, which yields $q(\mathbf{x}_\tau^{[T]} | \mathbf{x}_0^{[T]}) = \mathcal{N}(\mathbf{x}_\tau^{[T]}; \mathbf{x}_r + \sqrt{\bar{\alpha}_\tau}(\mathbf{x}_0^{[T]} - \mathbf{x}_r), (1 - \bar{\alpha}_\tau)\mathbf{I})$. The proof then generally follows [18] but with all latent variables in [18] being replaced by $\mathbf{x}_\tau^{[T]} - \mathbf{x}_r^{[T]}$. Then the terms in the VLB are

given by,

$$\mathcal{L}_{\tau-1} = D_{\text{KL}}(q(\mathbf{x}_{\tau-1}^{[T]} \mid \mathbf{x}_\tau^{[T]}, \mathbf{x}_0^{[T]}) \| p_\theta(\mathbf{x}_{\tau-1}^{[T]} \mid \mathbf{x}_\tau^{[T]})), \tag{40}$$

$$= \mathbb{E}_{\mathbf{x}_0^{[T]}, \boldsymbol{\epsilon}} \left[ \frac{1}{2\sigma_\tau^2} \left\| \mathbf{x}_r^{[T]} + \frac{1}{\sqrt{\alpha_\tau}} \left( \mathbf{x}_\tau^{[T]}(\mathbf{x}_0^{[T]}, \boldsymbol{\epsilon}, \mathbf{x}_c^{[T_c]}) - \mathbf{x}_r^{[T]} - \frac{\beta_\tau}{\sqrt{1-\bar{\alpha}_\tau}} \boldsymbol{\epsilon} \right) - \boldsymbol{\mu}_\theta(\mathbf{x}_\tau^{[T]}, \tau) \right\|^2 \right], \tag{41}$$

$$= \mathbb{E}_{\mathbf{x}_0^{[T]}, \boldsymbol{\epsilon}} \left[ \frac{1}{2\sigma_\tau^2} \left\| \mathbf{x}_r^{[T]} + \frac{1}{\sqrt{\alpha_\tau}} \left( \mathbf{x}_\tau^{[T]} - \mathbf{x}_r^{[T]} - \frac{\beta_\tau}{\sqrt{1-\bar{\alpha}_\tau}} \boldsymbol{\epsilon} \right) \right. \right.$$
$$\left. \left. - \mathbf{x}_r^{[T]} - \frac{1}{\sqrt{\alpha_\tau}} \left( \mathbf{x}_\tau^{[T]} - \mathbf{x}_r^{[T]} - \frac{\beta_\tau}{1-\bar{\alpha}_\tau} \boldsymbol{\epsilon}_\theta(\mathbf{x}_\tau^{[T]}, \mathbf{x}_c^{[T_c]}, \tau) \right) \right\|^2 \right], \tag{42}$$

$$= \mathbb{E}_{\mathbf{x}_0^{[T]}, \boldsymbol{\epsilon}} \left[ \frac{\beta_\tau^2}{2\sigma_\tau^2 \alpha_\tau (1-\bar{\alpha}_\tau)} \| \boldsymbol{\epsilon} - \boldsymbol{\epsilon}_\theta(\mathbf{x}_\tau^{[T]}, \mathbf{x}_c^{[T_c]}, \tau) \|^2 \right], \tag{43}$$

which is equivalent to Eq. 39 up to certain re-weighting factors. For $\mathcal{L}_\mathcal{T} = D_{\text{KL}}(q(\mathbf{x}_\mathcal{T}^{[T]} \mid \mathbf{x}_0^{[T]}) \| p(\mathbf{x}_\mathcal{T}^{[T]}))$, it is does not contribute to the gradient since it is irrelevant to $\boldsymbol{\theta}$, and $\mathbf{x}_r$ is also cancelled out in computing the KL, thus stopping the gradient from passing to $\boldsymbol{\eta}$ and $\boldsymbol{\gamma}$. $\square$

Analogous to the unconditional case, we have the following proposition, indicating that if the base distribution is SE(3)-equivariant and the transition kernel is SE(3)-equivariant, then the marginal is also SE(3)-equivariant.

**Proposition A.3.** *If the base distribution $p_\mathcal{T}(\mathbf{x}_\mathcal{T}^{[T]} \mid \mathbf{x}_c^{[T_c]})$ is SE(3)-equivariant and the transition kernels $p_{\tau-1}(\mathbf{x}_{\tau-1}^{[T]} \mid \mathbf{x}_\tau^{[T]}, \mathbf{x}_c^{[T_c]})$ of all diffusion steps $\tau \in \{1, \cdots, \mathcal{T}\}$ are SE(3)-equivariant, then the marginal[6] $p_\tau(\mathbf{x}_\tau^{[T]} \mid \mathbf{x}_c^{[T_c]})$ at any diffusion step $\tau \in \{0, \cdots, \mathcal{T}\}$ is SE(3)-equivariant.*

*Proof.* The proof is similarly given by induction.

**Induction base.** When $\tau = \mathcal{T}$, the distribution is the base distribution $p_\mathcal{T}(\mathbf{x}_\mathcal{T}^{[T]} \mid \mathbf{x}_c^{[T_c]})$ is SE(3)-equivariant, as it is designed.

**Induction step.** Suppose the marginal at diffusion step $\tau$, *i.e.*, $p_\tau(\mathbf{x}_\tau^{[T]} \mid \mathbf{x}_c^{[T_c]})$, is SE(3)-equivariant, then we have

$$p_{\tau-1}(\mathbf{R}\mathbf{x}_{\tau-1}^{[T]} + \mathbf{r} \mid \mathbf{R}\mathbf{x}_c^{[T_c]} + \mathbf{r}) \tag{44}$$

$$= \int p_{\tau-1}(\mathbf{R}\mathbf{x}_{\tau-1}^{[T]} + \mathbf{r} \mid \mathbf{x}_\tau^{[T]}, \mathbf{R}\mathbf{x}_c^{[T_c]} + \mathbf{r}) p_\tau(\mathbf{x}_\tau^{[T]} \mid \mathbf{R}\mathbf{x}_c^{[T_c]} + \mathbf{r}) \mathrm{d}\mathbf{x}_\tau^{[T]}, \tag{45}$$

$$= \int p_{\tau-1}(\mathbf{R}\mathbf{x}_{\tau-1}^{[T]} + \mathbf{r} \mid \mathbf{R}(\mathbf{R}^{-1}(\mathbf{x}_\tau^{[T]} - \mathbf{r})) + \mathbf{r}, \mathbf{R}\mathbf{x}_c^{[T_c]} + \mathbf{r}) p_\tau(\mathbf{R}(\mathbf{R}^{-1}(\mathbf{x}_\tau^{[T]} - \mathbf{r})) + \mathbf{r} \mid \mathbf{R}\mathbf{x}_c^{[T_c]} + \mathbf{r}) \mathrm{d}\mathbf{x}_\tau^{[T]}, \tag{46}$$

$$= \int p_{\tau-1}(\mathbf{x}_{\tau-1}^{[T]} \mid \mathbf{R}^{-1}(\mathbf{x}_\tau^{[T]} - \mathbf{r}), \mathbf{x}_c^{[T_c]}) p_\tau(\mathbf{R}^{-1}(\mathbf{x}_\tau^{[T]} - \mathbf{r}) \mid \mathbf{x}_c^{[T_c]}) \mathrm{d}\mathbf{x}_\tau^{[T]}, \tag{47}$$

$$= \int p_{\tau-1}(\mathbf{x}_{\tau-1}^{[T]} \mid \mathbf{y}_\tau^{[T]}, \mathbf{x}_c^{[T_c]}) p_\tau(\mathbf{y}_\tau^{[T]} \mid \mathbf{x}_c^{[T_c]}) \det(\mathbf{R}) \mathrm{d}\mathbf{y}_\tau^{[T]}, \tag{48}$$

$$= p_{\tau-1}(\mathbf{x}_{\tau-1}^{[T]} \mid \mathbf{x}_c^{[T_c]}), \tag{49}$$

which concludes the proof. $\square$

### A.3 Optimizable Equivariant Prior

**Theorem A.4.** *The prior implemented by the parameterization in Eq. 9, 10, and 11 subsumes CoM-based priors and fixed point-wise priors.*

---

[6]Here the marginal refers to marginalizing the intermediate states in previous diffusion steps, while still being conditional on the input condition $\mathbf{x}_c^{[T_c]}$.

*Proof.* We repeat the parameterizations specified by Eq. 9, 10, and 11 below for better readability.

$$\mathbf{x}_r^{(t)} = \sum_{s \in [T_c]} \mathbf{w}^{(t,s)} \hat{\mathbf{x}}_c^{(s)}, \quad \text{s.t.} \sum_{s \in [T_c]} \mathbf{w}^{(t,s)} = \mathbf{1}_N, \tag{50}$$

$$\mathbf{W}_{t,s} = [\boldsymbol{\gamma} \otimes \hat{\mathbf{h}}_c^{[T_c]}]_{t,s} \in \mathbb{R}^N, \tag{51}$$

$$\mathbf{w}^{(t,s)} = \begin{cases} \mathbf{W}_{t,s} & s < T_c - 1, \\ \mathbf{1}_N - \sum_{s=0}^{T_c-2} \mathbf{W}_{t,s} & s = T_c - 1. \end{cases} \tag{52}$$

We first show $\mathbf{x}_r^{[T]}$ can reduce to the CoM-based priors. Let $\hat{\mathbf{x}}_c^{(s)} = \text{CoM}(\mathbf{x}_c^{(s)})$, $\hat{\mathbf{h}}_c^{(s)} = \frac{1}{T_c}\mathbf{1}_N$, $\gamma^{(t)} = 1$. In this case,

$$\mathbf{x}_r^{(t)} = \sum_{s \in [T_c]} \mathbf{w}^{(t,s)} \hat{\mathbf{x}}_c^{(s)}, \tag{53}$$

$$= \sum_{s \in [T_c-1]} \gamma^{(t)} \frac{1}{T_c} \mathbf{1}_N \text{CoM}(\mathbf{x}_c^{(s)}) + (\mathbf{1}_N - \sum_{s \in [T_c-1]} \gamma^{(t)} \frac{1}{T_c} \mathbf{1}_N)\text{CoM}(\mathbf{x}_c^{(T_c-1)}), \tag{54}$$

$$= \sum_{s \in [T_c-1]} \frac{1}{T_c} \mathbf{1}_N \text{CoM}(\mathbf{x}_c^{(s)}) + (\mathbf{1}_N - \frac{T_c-1}{T_c} \mathbf{1}_N)\text{CoM}(\mathbf{x}_c^{(T_c-1)}), \tag{55}$$

$$= \sum_{s \in [T_c-1]} \frac{1}{T_c} \mathbf{1}_N \text{CoM}(\mathbf{x}_c^{(s)}) + \frac{1}{T_c} \mathbf{1}_N \text{CoM}(\mathbf{x}_c^{(T_c-1)}), \tag{56}$$

$$= \frac{1}{T_c} \sum_{s \in [T_c]} \text{CoM}(\mathbf{x}_c^{(s)}), \tag{57}$$

where $\frac{1}{T_c} \sum_{s \in [T_c]} \text{CoM}(\mathbf{x}_c^{(s)})$ is the generalization of the CoM-based priors [21, 13] in the multiple frame conditioning scenario, which reduces to $\text{CoM}(\mathbf{x}_c^{(0)})$ when $T_c = 1$.

To show $\mathbf{x}_r^{[T]}$ can reduce to fixed point-wise priors is straightforward. Let $\hat{\mathbf{x}}_c^{(s)} = \mathbf{x}_c^{(s)}$, $\hat{\mathbf{h}}^{[T_c]} = \text{Onehot}(s^*)\mathbf{1}_{T_c \times N}$ and $\gamma^{(t)} = 1, \forall t$. Then $\mathbf{w}^{(t,s)} = \text{Onehot}(s^*)\mathbf{1}_{T_c \times N}$. Therefore,

$$\mathbf{x}_r^{(t)} = \sum_{s \in [T_c]} \mathbf{w}^{(t,s)} \mathbf{x}_c^{(s)}, \tag{58}$$

$$= \sum_{s \in [T_c]} \text{Onehot}(s^*)\mathbf{1}_{T_c \times N} \mathbf{x}_c^{(s)}, \tag{59}$$

$$= \mathbf{x}_c^{(s^*)}, \tag{60}$$

where $\mathbf{x}_c^{(s^*)}$ is the point-wise equivariant prior, and $s^* \in [T_c]$ is the frame index in the conditioning trajectory for this specific prior. $\square$

We also provide an illustrative comparison of these equivariant priors in Fig. 3.

### A.4 Proof of Theorem 4.1

**Theorem 4.1** (SE(3)-equivariance of EGTN). *Let* $\mathbf{x}'^{[T]}, \mathbf{h}'^{[T]} = f_{\text{EGTN}}\left(\mathbf{x}^{[T]}, \mathbf{h}^{[T]}, \mathcal{E}\right)$. *Then we have* $g \cdot \mathbf{x}'^{[T]}, \mathbf{h}'^{[T]} = f_{\text{EGTN}}\left(g \cdot \mathbf{x}^{[T]}, \mathbf{h}^{[T]}, \mathcal{E}\right), \forall g \in SE(3)$.

*Proof.* $f_{\text{EGTN}}$ is a stack of $L$ EGNN and temporal attention layer in alternated fashion, formally written as $f_{\text{EGTN}} = \underbrace{f_{\text{attn}} \circ f_{\text{EGNN}} \circ \cdots \circ f_{\text{attn}} \circ f_{\text{EGNN}}}_{L \times (f_{\text{attn}} \circ f_{\text{EGNN}})}$. Since the chain of SE(3)-equivariant function is also SE(3)-equivariant, it suffices to prove $f_{\text{attn}}$ is SE(3)-equivariant, in that the SE(3)-equivariance of EGNN directly follows [43].

It is directly verified that the attention coefficients $\mathbf{a}^{(t,s)} \in \mathbb{R}$ in Eq. 5 and the query $\mathbf{q}^{[T]}$, key $\mathbf{k}^{[T]}$, and value $\mathbf{v}^{[T]}$ are all SE(3)-invariant, since they are derived based on the SE(3)-invariant input

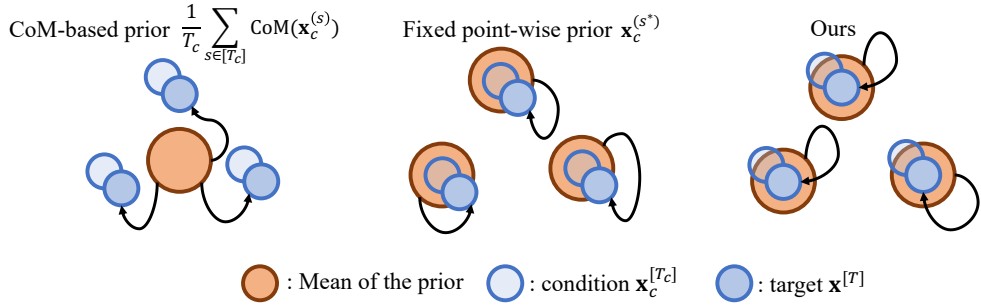

$\bigcirc$ : Mean of the prior   $\bigcirc$ : condition $\mathbf{x}_c^{[T_c]}$   $\bigcirc$ : target $\mathbf{x}^{[T]}$

Figure 3: An illustration of different equivariant priors. For simplicity in the chart here we only illustrate the case when $N = 3$ and $T_c = 1, T = 1$.

$\mathbf{h}^{[T]}$. This directly leads to the SE(3)-invariance of the updated node feature $\mathbf{h}'^{[T]}$. For the updated coordinates,

$$\mathbf{x}_{\mathrm{tr}}'^{(t)} = \mathbf{x}_{\mathrm{tr}}^{(t)} + \sum_{s \in [T]} \mathbf{a}_{\mathrm{tr}}^{(t,s)} \varphi_{\mathbf{x}}(\mathbf{v}_{\mathrm{tr}}^{(t,s)})(\mathbf{x}_{\mathrm{tr}}^{(t)} - \mathbf{x}_{\mathrm{tr}}^{(s)}), \tag{61}$$

$$= \mathbf{R}\mathbf{x}^{(t)} + \mathbf{r} + \sum_{s \in [T]} \mathbf{a}^{(t,s)} \varphi_{\mathbf{x}}(\mathbf{v}^{(t,s)})(\mathbf{R}\mathbf{x}^{(t)} + \mathbf{r} - \mathbf{R}\mathbf{x}^{(s)} - \mathbf{r}), \tag{62}$$

$$= \mathbf{R}\mathbf{x}^{(t)} + \mathbf{r} + \mathbf{R}\left(\sum_{s \in [T]} \mathbf{a}^{(t,s)} \varphi_{\mathbf{x}}(\mathbf{v}^{(t,s)})(\mathbf{x}^{(t)} - \mathbf{x}^{(s)})\right), \tag{63}$$

$$= \mathbf{R}\left(\mathbf{x}^{(t)} + \sum_{s \in [T]} \mathbf{a}^{(t,s)} \varphi_{\mathbf{x}}(\mathbf{v}^{(t,s)})(\mathbf{x}^{(t)} - \mathbf{x}^{(s)})\right) + \mathbf{r}, \tag{64}$$

$$= \mathbf{R}\mathbf{x}'^{(t)} + \mathbf{r}, \tag{65}$$

where the variables with subscript $\mathrm{tr}$ refers to their transformed counterparts when the input $\mathbf{x}^{[T]}$ is transformed into $\mathbf{R}\mathbf{x}^{[T]} + \mathbf{r}$. Thus it completes the proof of SE(3)-equivariance of the temporal attention $f_{\mathrm{attn}}$ and hence the entire $f_{\mathrm{EGTN}}$.

$\square$

# B   More Details on Experiments

## B.1   Compute Resources

We use Distributed Data Parallel on 4 Nvidia A6000 GPUs to train all the models. The training on NBody and ETH-UCY take around 12 hours while each MD17 training phase takes about a day. Our CPUs were standard intel CPUs.

## B.2   Hyper-parameters

We provide the detailed hyper-parameters of GeoTDM in Table 7. We adopt Adam optimizer with betas $(0.9, 0.999)$ and $\epsilon = 10^{-8}$. For all experiments, we use the linear noise schedule per [18] with $\beta_{\mathrm{start}} = 0.02$ and $\beta_{\mathrm{end}} = 0.0001$.

Table 7: Hyper-parameters of GeoTDM in the experiments.

|  | n_layer | hidden | time_emb_dim | $\mathcal{T}$ | batch_size | learning_rate |
|---|---|---|---|---|---|---|
| N-body | 6 | 128 | 32 | 1000 | 128 | 0.0001 |
| MD | 6 | 128 | 32 | 1000 | 128 | 0.0001 |
| ETH | 4 | 64 | 32 | 100 | 100 | 0.0005 |

## B.3 Baselines

For the frame-to-frame prediction models, including RF [27], EGNN [43], TFN [56], and SE(3)-Transformer [9], we adopt the implementation in the codebase maintained by [43]. To yield a strong comparison, instead of taking one frame as input to directly predict the final frame, we employ a discretized NeuralODE [4]-style training and inference procedure. In particular, we train the models with position $\mathbf{x}^{(t)}$ and velocity (computed as the difference of the current and previous frame, *i.e.*, $\mathbf{v}^{(t)} = \mathbf{x}^{(t)} - \mathbf{x}^{(t-1)}$) as input to predict the next velocity $\hat{\mathbf{v}}^{(t+1)}$. The position for the next step is integrated as $\hat{\mathbf{x}}^{(t+1)} = \mathbf{x}^{(t)} + \hat{\mathbf{v}}^{(t+1)}$. The training loss is computed as the Mean Squared Error (MSE) between the predicted position $\hat{\mathbf{x}}^{(t+1)}$ and the ground truth position $\mathbf{x}_{\text{gt}}^{(t+1)}$. In inference time, a roll-out prediction is conducted, which iteratively predict the next step by feeding the predicted position and velocity at the current step, for a total of $T$ steps. We follow the hyper-parameter tuning guideline for these baselines by [16] which conduct a random search over the space spanned by the number of layers in $\{2, 4, 6, 8\}$, the hidden dimension $\{32, 64, 128\}$, learning rate $\{5e-3, 1e-3, 5e-4, 1e-4\}$, and batch size $32, 64, 128, 256$, and select the model with best performance. All models are trained towards convergence with an early-stopping counter of 5, with validation performed every 20 epochs.

For EqMotion, we directly adopt the code by [62] and their suggested hyper-parameters for the N-body datasets and MD17 datasets.

For SVAE [67] and SGAN [14], these methods are originally developed for the pedestrian trajectory forecasting task. The backbone model that processes the input trajectory consists of social pooling operation and GRU or LSTM blocks for temporal processing. In order the make them favorable in tackling geometric systems which additionally include node features and edge features, we replace the social pooling operations by MPNNs [11] in both encoder (or discriminator) and decoder (or generator) to synthesize the information on the geometric graph. The temporal module is still kept as GRU for SVAE and LSTM for SGAN, following their original implementations. We also search over the best hyper-parameters which additionally involve the KL-divergence weight in $\{1, 0.1, 0.01, 0.001\}$ for SVAE according to the validation ELBO. For EGVAE, we replace the MPNNs in SVAE by EGNN [43], and restructured the latent space of the prior with both equivariant and invariant latent features. By this means, EGVAE is also guaranteed to model an equivariant distribution in the conditional case and an invariant distribution in the unconditional case.

## B.4 Model

In detail, the EGCL layer [43] is given by:

$$\mathbf{m}_{ij} = \varphi_{\mathbf{m}}\left(\mathbf{h}_i, \mathbf{h}_j, \|\mathbf{x}_i - \mathbf{x}_j\|, \mathbf{e}_{ij}\right), \tag{66}$$

$$\mathbf{h}'_i = \varphi_{\mathbf{h}}\left(\mathbf{h}_i, \sum_{j \in \mathcal{N}(i)} \mathbf{m}_{ij}\right), \tag{67}$$

$$\mathbf{x}'_i = \mathbf{x}_i + \sum_{j \in \mathcal{N}(i)} \varphi_{\mathbf{x}}\left(\mathbf{m}_{ij}\right)\left(\mathbf{x}_i - \mathbf{x}_j\right), \tag{68}$$

where $\varphi_{\mathbf{m}}$, $\varphi_{\mathbf{h}}$, and $\varphi_{\mathbf{x}}$ are all MLPs. We also provide a schematic of our proposed EGTN in Fig. 4 for better illustration.

## B.5 Evaluation Metrics in the Unconditional Case

All these metrics are evaluated on a set of model samples with the same size as the testing set.

**Marginal score** is computed as the absolute difference of two empirical probability density functions. Practically, we collect the $x, y, z$ coordinates at each time step marginalized over all nodes in all systems in the predictions and the ground truth (testing set). Then we split the collection into 50 bins and compute the MAE in each bin, finally averaged across all time steps to obtain the score. Note that on MD17, instead of computing the pdf on coordinates, we compute the pdf on the length of the chemical bonds, which is a clearer signal that correlates to the validity of the generated MD trajectory, since during MD simulation the bond lengths are usually stable with very small vibrations. Marginal score gives a broad statistical measurement how each dimension of the generated samples align with the original data.

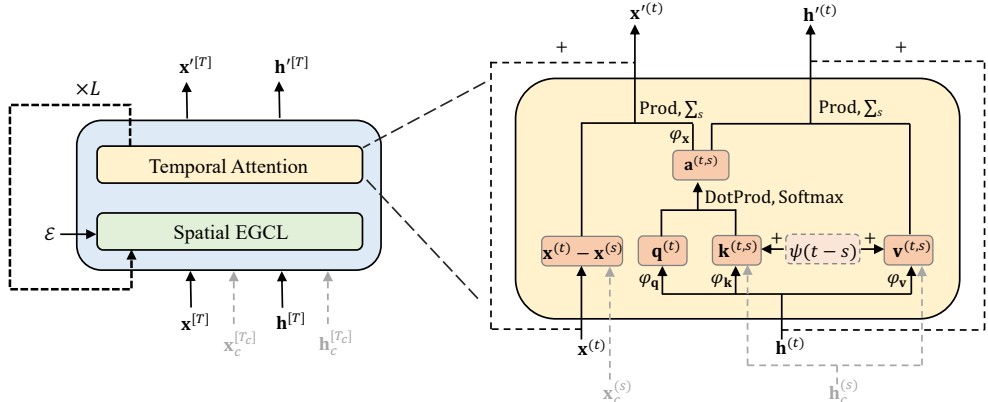

Figure 4: Schematic of the proposed EGTN, which alternates the EGCL layer for extracting spatial interactions and the temporal attention layer for modeling temporal sequence. Additional conditional information $\mathbf{x}_c^{[T_c]}$ and $\mathbf{h}_c^{[T_c]}$ can also be processed using cross-attention. The relative temporal embedding $\psi(t - s)$ is added to the key and value. DotProd refers to dot product and Softmax is performed over indexes of $s$.

**Classification score** is computed as the cross-entropy loss of a sequence classification model that aims to distinguish whether the trajectory is generated by the model or from the testing set. To be specific, we construct a dataset mixed by the generated samples and the testing set, and randomly split it into 80% and 20% subsets for training and testing. Then the model is trained on the training set and the classification score is computed as the cross-entropy on the testing set. We use a 1-layer EqMotion with a classification head as the model. The classification score provided intuition on how difficult it is to distinguish the generated samples and the original data.

**Prediction score** is computed as the MSE loss of a train-on-synthetic-test-on-real sequence to sequence model. In detail, we train a 1-layer EqMotion on the sampled dataset with the task of predicting the second half of the trajectory given the first half. We then evaluate the model on the testing set and report the MSE as the prediction score. Prediction score provides intuition on the capability of the generative model on generating synthetic data that well aligns with the ground truth.

## C  More Experiments and Discussions

### C.1  Model Composition for Longer Trajectory

Since attention is utilized to extract temporal information, the time complexity scales quadratically with the length of the input trajectory, both during training and inference. In practice, we can instead train models on shorter trajectories and compose them during inference for longer trajectories, in both unconditional and conditional cases. For target trajectories with length $T$, we can first decompose it into $K$ several equal-length[7] non-overlapping intervals with time span $\Delta T$. Then, for the unconditional case, we have

$$p(\mathbf{x}^{[T]}) = p(\mathbf{x}^{[\Delta T]}) \prod_{k=1}^{K-1} p(\mathbf{x}^{k\Delta T + [\Delta T]} \mid \mathbf{x}^{(k-1)\Delta T + [\Delta T]}), \tag{69}$$

by assuming mild conditional independence, where $p(\mathbf{x}^{[\Delta T]})$ is an unconditional model for trajectory with length $\Delta T$, and $p(\mathbf{x}^{k\Delta T + [\Delta T]} \mid \mathbf{x}^{(k-1)\Delta T + [\Delta T]})$ can be learned by a conditional model for short trajectories. The conditional case directly follows by factorizing into products of conditional distribution over shorter trajectories.

We provide a demonstration of such technique as gifs in the **supplementary file**.

---

[7]In fact they do not necessarily need to be equal-length. Here we make such assumption for conciseness of the presentation.

Table 8: The effect of diffusion steps in the unconditional generation setting (top) and conditional forecasting setting (bottom).

| | | Aspirin | | | Charged Particle | |
|---|---|---|---|---|---|---|
| | Marginal ↓ | Classification ↑ | Prediction ↓ | Marginal ↓ | Classification ↑ | Prediction ↓ |
| $\mathcal{T} = 100$ | 0.808 | 0.0242 | 0.0243 | 0.0065 | 0.170 | 0.0118 |
| $\mathcal{T} = 1000$ | 0.726 | 0.0348 | 0.0212 | 0.0055 | 0.556 | 0.00978 |
| | | Aspirin | | | Charged Particle | |
| | ADE ↓ | FDE ↓ | NLL ↓ | ADE ↓ | FDE ↓ | NLL ↓ |
| $\mathcal{T} = 100$ | 0.110 | 0.198 | -2125.7 | 0.120 | 0.280 | -547.5 |
| $\mathcal{T} = 1000$ | 0.107 | 0.193 | -3461.4 | 0.110 | 0.258 | -982.7 |

## C.2 Number of Diffusion Steps

We provide results in the unconditional generation setting for $\mathcal{T} = 100$. The results are in Table 8. Compared with the conditional setting, the unconditional generation is more challenging in that is needs to generate trajectories without any given reference geometries. We observe a drop in performance when $\mathcal{T}$ is decreased from 1000 to 100. However, the performance with only 100 diffusion steps is still significantly better than SVAE.

## C.3 Sampling Time

In the table below we display the generation metrics and the inference time per batch with batch size 128 on MD17 Aspirin molecule. We compare GeoTDM with EGVAE, an autoregressive VAE-based method with EGNN as the backbone. Here GeoTDM-100 and GeoTDM-1000 refer to GeoTDM using 100 and 1000 diffusion steps, respectively.

Table 9: Sampling runtime comparison on MD17 Aspirin molecule.

| | Marginal | Classification | Prediction | Time per batch |
|---|---|---|---|---|
| EGVAE | 2.650 | $1.31 \times 10^{-4}$ | 0.0386 | 0.6±0.1 |
| GeoTDM-100 | 0.808 | $2.42 \times 10^{-2}$ | 0.0243 | 7.9±0.8 |
| GeoTDM-1000 | 0.726 | $3.48 \times 10^{-2}$ | 0.0212 | 74.2±2.1 |

We observe that GeoTDM-100 is approximately 10 times slower than EGVAE, since the model requires 100 calls of the denoising network to generate one batch, while EGVAE consumes the same number of calls as the length of the trajectory (20 in this case) due to autoregressive modeling. Although GeoTDM is slower, the gain in performance is significant and the quality of the generated trajectory is remarkably better than that of EGVAE. When further increasing the number of diffusion steps to 1000, the performance becomes better while requiring much more compute.

However, it is worth noticing that all these deep learning-based methods are significantly faster than traditional methods like DFT, which typically requires hours to even several days to converge depending on the scale of the system, according to OCP [3]. Therefore, although GeoTDM becomes slower than VAEs when using larger number of diffusion steps, it is still much faster than DFT, which indicates its practical value in generating geometric trajectories like molecular dynamics simulation.

The computation overhead of diffusion models compared with VAEs or GANs has been a well-known issue. We recognize enhancing the efficiency of GeoTDM as an interesting direction of future work, potentially through adopting faster solvers like DDIM [50] or DPMSolver [29], performing consistency distillation [51], or developing latent diffusion models [64] that take advantage of a more compact representation of the spatio-temporal geometric space.

## C.4 Standard Deviations

We provide the standard deviations in Table 10 and 11.

Table 10: Conditional generation results of GeoTDM on N-body charged particle, spring, and gravity. Results (mean ± standard deviation) are computed from 5 samples.

| | Particle | | Spring | | Gravity | |
|---|---|---|---|---|---|---|
| | ADE | FDE | ADE | FDE | ADE | FDE |
| GeoTDM | 0.110±0.014 | 0.258±0.032 | 0.0030±0.0004 | 0.0079±0.0010 | 0.256±0.015 | 0.613±0.034 |
| SVAE | 0.378±0.005 | 0.732±0.005 | 0.0120±0.0003 | 0.0209±0.0004 | 0.582±0.007 | 1.101±0.015 |

Table 11: Conditional generation results of GeoTDM on MD17. Results (mean ± standard deviation) are computed from 5 samples.

| Aspirin | | Benzene | | Ethanol | | Malonaldehyde | |
|---|---|---|---|---|---|---|---|
| ADE | FDE | ADE | FDE | ADE | FDE | ADE | FDE |
| 0.107±0.005 | 0.193±0.016 | 0.023±0.001 | 0.039±0.004 | 0.115±0.012 | 0.209±0.035 | 0.107±0.010 | 0.176±0.025 |

| Naphthalene | | Salicylic | | Toluene | | Uracil | |
|---|---|---|---|---|---|---|---|
| ADE | FDE | ADE | FDE | ADE | FDE | ADE | FDE |
| 0.064±0.002 | 0.087±0.007 | 0.083±0.004 | 0.120±0.012 | 0.083±0.004 | 0.121±0.011 | 0.074±0.003 | 0.099±0.009 |

## C.5 More Discussions with Existing Works

Below we discuss the unique challenges for designing GeoTDM compared with MID [12] and geometric diffusion models like GeoDiff [66] and GeoLDM [64], and how we tackle these challenges.

**Modeling geometric trajectories.** Although MID can model trajectories, it leverages Trajectron++ [40] backbone which takes as input the position vectors through a Transformer network. It requires non-trivial effort to incorporate additional node features and edge features into MID, while for GeoTDM, we design a general backbone EGTN that can process geometric trajectories while preserving equivariance. Existing geometric diffusion models (*e.g.*, GeoDiff and GeoLDM) never consider modeling the temporal dynamics and their backbone can only work on static (single-frame) geometric strctures.

**Incorporating equivariance into temporal diffusion.** While geometric diffusion models have discussed proper ways to inject equivariance into diffusion models, it is unclear how to preserve equivariance when each hidden variable in the diffusion process has an additional dimension of time. In this work, we formally define equivariance constraint we want to impose on the marginal distribution, and how to design the prior and transition kernel in order to fulfill the constraint, in the context where all hidden variables are geometric trajectories. This is technically very different from existing works (*e.g.*, GeoDiff and GeoLDM) since the dimension of the data is fundamentally different, which leads to different analyses.

**Consideration of both conditional and unconditional generation scenarios.** MID is only designed and evaluated in the conditional setting where the task is to forecast the future trajectory given initial frames. GeoDiff and GeoLDM only operate in the unconditional setting where the task is to generate the structure without any initial 3D structure information. In this work, we systematically discuss both unconditional and conditional generation for geometric trajectories, and elaborate on how to design the prior and transition kernel to meet the equivariance constraint.

**Parameterization of the learnable equivariant prior.** In the conditional case, we propose to parameterize the equivariant prior with a lightweight EGTN. Such approach offers more flexibility in the equivariant prior, enabling optimizing it during training, which is also proved to be able to subsume existing center-of-mass (CoM) based parameterization (see Theorem A.4 in Appendix). Experiments in ablation studies also verify the superiority of such design.

We summarize the points above in Table 12.

## D More Visualizations

We provide more visualizations in Fig. 5, 6, 8, 7, 9, and 10. Please refer to their captions for the detailed descriptions.

Table 12: Technical differences between GeoTDM and existing works.

| | Trajectory | Equivariance | Conditional | Unconditional | Learnable Prior |
|---|---|---|---|---|---|
| MID [12] | ✓ | | ✓ | | |
| GeoDiff [66], GeoLDM [64] | | ✓ | | ✓ | |
| Our GeoTDM | ✓ | ✓ | ✓ | ✓ | ✓ |

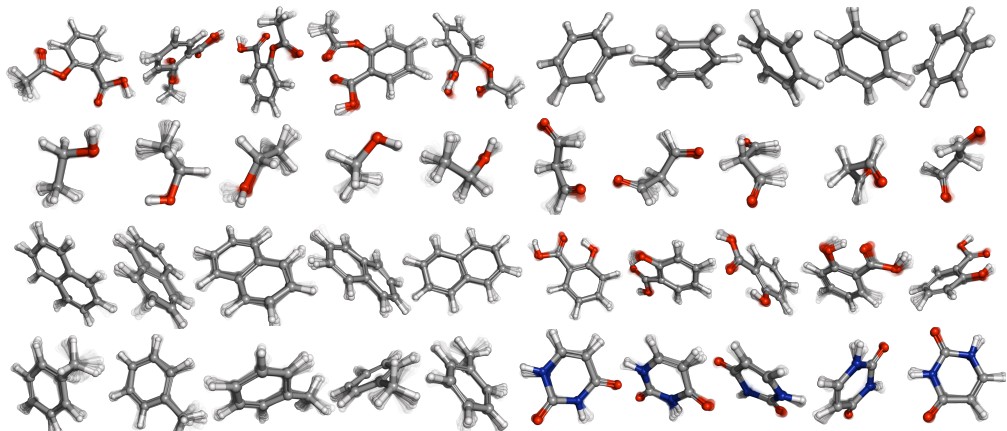

Figure 5: Uncurated samples of GeoTDM on MD17 dataset in the unconditional generation setup. From top-left to bottom-right are trajectories of the eight molecules: Aspirin, Benzene, Ethanol, Malonaldehyde, Naphthalene, Salicylic, Toluene, and Uracil. Five samples are displayed for each molecule. GeoTDM generates high quality samples. It well captures the vibrations and rotating behavior of the methyl groups in Aspirin and Ethanol. The bonds on the benzene ring are also more stable, aligning with findings in chemistry.

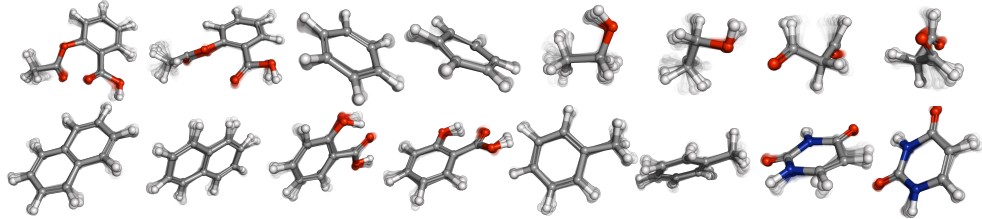

Figure 6: Samples from MD17 dataset.

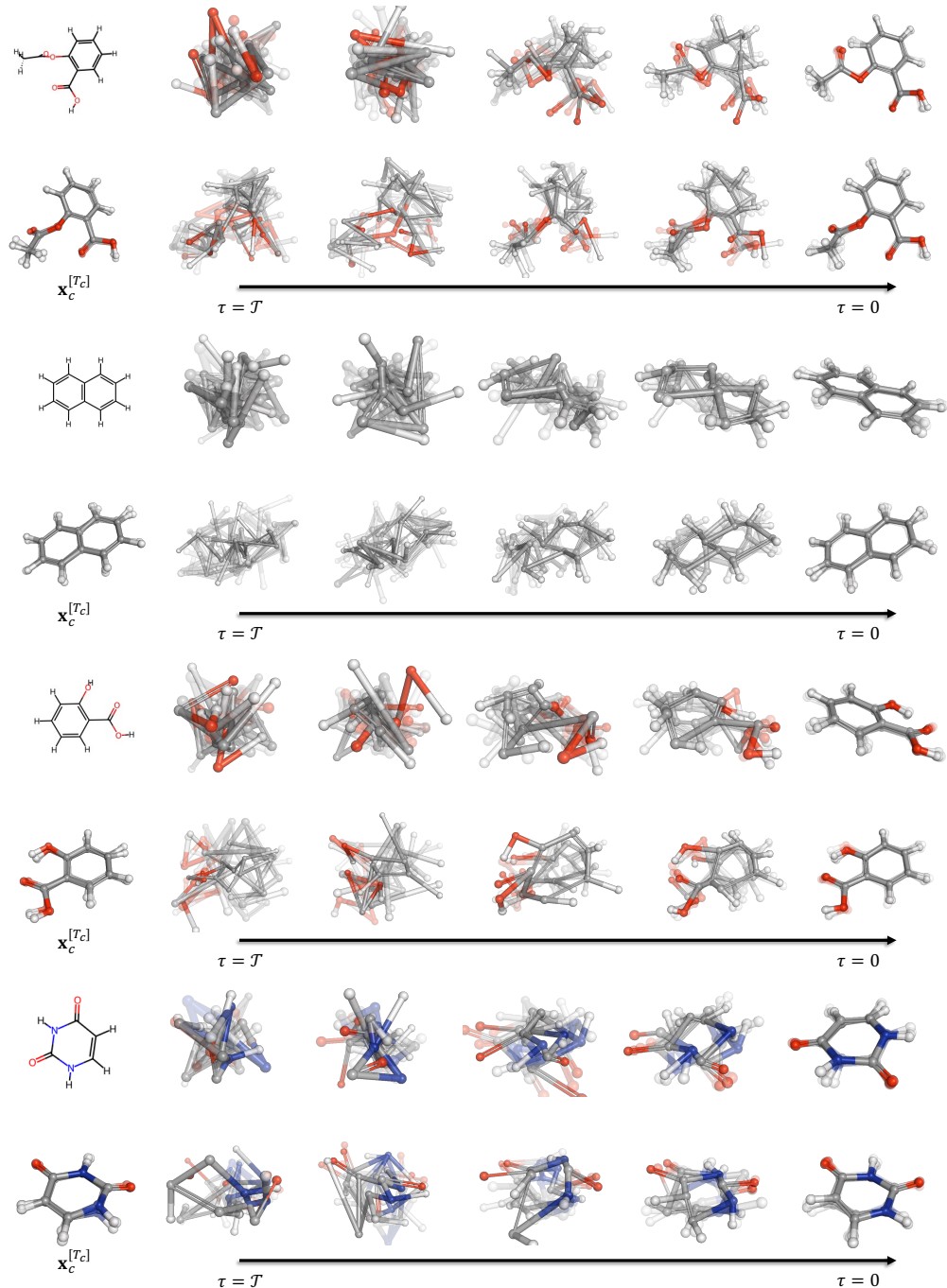

Figure 7: Visualization of the diffusion trajectory at different diffusion steps. From top to bottom: *Aspirin*, *Naphthalene*, *Salicylic*, *Uracil*. For each molecule, the first row shows the unconditional generation process, where the model generates the trajectory from the **invariant prior** purely from the molecule graph without any conditioning structure. The second row refers to the conditional generation, where the model generates from the **equivariant prior**, conditioning on some given frames $\mathbf{x}_c^{[T_c]}$. Notably, the equivariant prior (see samples at $\tau = \mathcal{T}$ in each second row) preserves some structural information encapsulated in $\mathbf{x}_c^{[T_c]}$, thanks to our flexible parameterization.

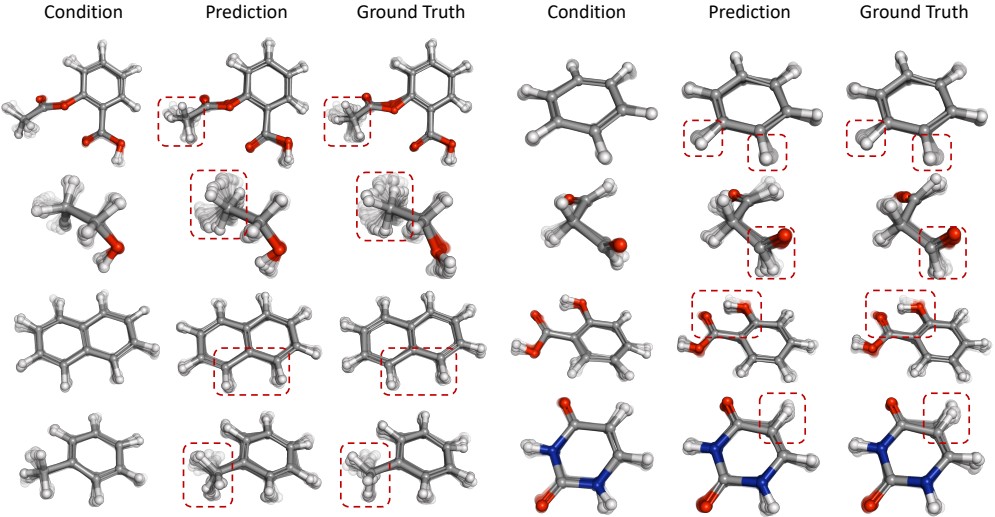

Figure 8: Uncurated samples of GeoTDM on MD17 dataset in the conditional forecasting setting. We highlight some regions of interest in red dashed boxes. GeoTDM delivers samples with very high accuracy while also capturing some stochasticity of the molecular dynamics.

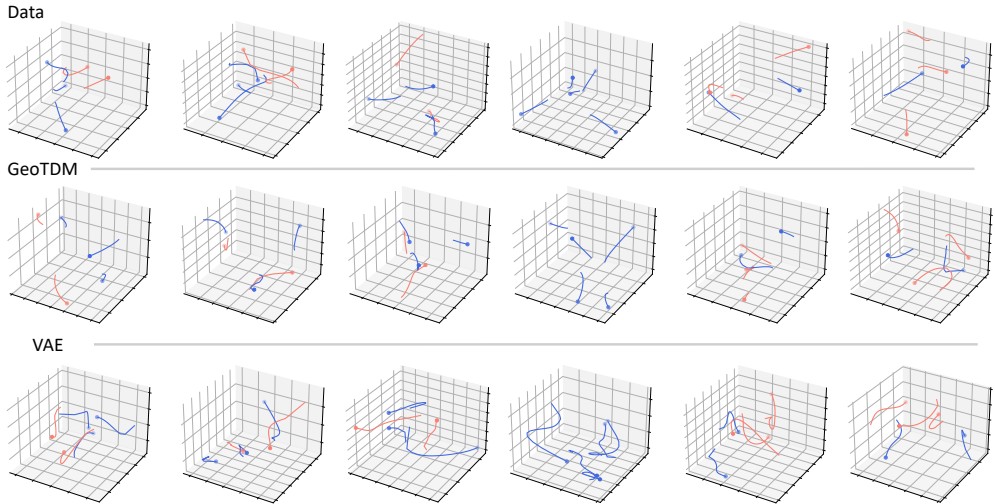

Figure 9: Visualization of data samples and generated samples by GeoTDM and SVAE in the unconditional setting on Charged Particles dataset. Nodes with color red and blue have the charge of +1/-1, respectively. Best viewed by zooming in.

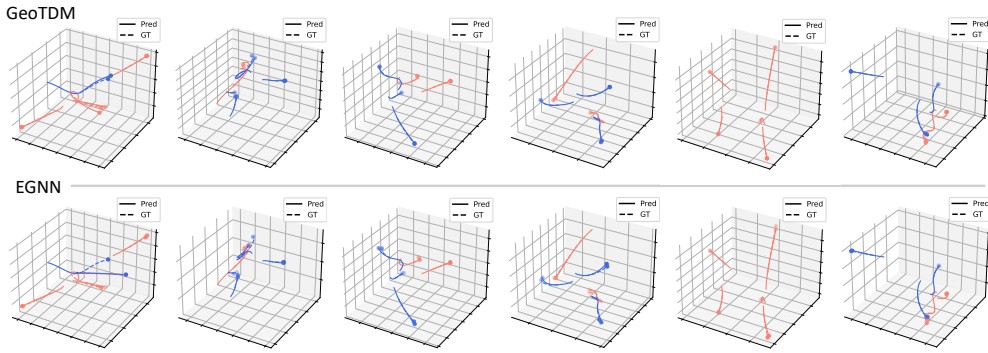

Figure 10: Visualization of predictions by GeoTDM and EGNN in the conditional setting on Charged Particles dataset. Nodes with color red and blue have the charge of +1/-1, respectively. Best viewed by zooming in.

