# OpenReview forum: "Geometric Trajectory Diffusion Models"
_NeurIPS.cc/2024/Conference — NeurIPS 2024 poster_

### Official Review · Reviewer_RWWk · 2024-07-05

**Soundness:** 3
**Presentation:** 3
**Contribution:** 3
**Rating:** 7
**Confidence:** 4

**Summary:**

A new diffusion-based generative model for modeling complex 3D geometric structures with time-evolving trajectories is proposed. By introducing the SE(3) equivariance property, temporal attention, and learnable geometric prior into the discretized diffusion model, the proposed model can achieve high performance in learning the distribution of geometric trajectories while preserving the symmetry of geometric systems. It can also be improved to attain conditional generation by using the equivariant cross-attention mechanism. Various types of experiments show high accuracy and performance of the proposed model in cases such as molecular dynamics and simulation.

**Strengths:**

1. The introduction and methods are well-written and clear.
2. The idea and methodology of introducing and processing the time sequence information are reasonable and well-defined with good novelty.  And the appendix provides sufficient proof and detailed analysis.
3. The paper explained and analyzed the case of conditional generation in detail.
4. The experimental setting and results are demonstrated in very extensive and detailed ways.
5. Clear ablation studies.

**Weaknesses:**

No major weakness. Please refer to the part of Questions for my concerns.

**Questions:**

1. As shown in the experiment part, the speed and efficiency are still the main limitations of the proposed model. Since the transition kernel and the prior are modified with extra restrictions, will it be difficult to redesign the framework (both forward and backward) into the continuous diffusion ODE/SDE form? (which might be important for further acceleration)

2. Except for the part of processing temporal information, could you compare more details about the proposed EGTN/EGCL with the E(3) Equivariant Diffusion Model in the paper [1]?

3. Why the geometric restrictions are also important in N-body simulation? What is the 'geometric structure' in these scenarios? It would be better if the author could provide more background knowledge about geometric trajectory and the physics simulation/molecular dynamics in the appendix.

4. Based on the visualization results in the appendix, the change in the molecular structure seems to be 'small'. Will such short trajectories be practically useful/meaningful in application?

[1]: Hoogeboom, Emiel, et al. "Equivariant diffusion for molecule generation in 3d." International conference on machine learning. PMLR, 2022.

**Limitations:**

The authors adequately addressed and discussed the limitations about computation. It would be better if the paper could also mention the current gap between the proposed model and the industrial/academic demand in medical/physical fields.

---

> ### Author Rebuttal · Authors · 2024-08-07
>
> Thank you for your constructive review and suggestions! We provide point-to-point response below.
>
> > **[Q1] Since the transition kernel and the prior are modified with extra restrictions, will it be difficult to redesign the framework (both forward and backward) into the continuous diffusion ODE/SDE form? (which might be important for further acceleration)**
>
> Thank you for the insightful comment! Interestingly, our framework can be naturally extended into continuous-time formulation with a simple reparameterization for the learnable prior. In detail, denote the learnable prior as $\mathbf{x}\_r^{[T]}$, then with simple change-of-variables $\mathbf{x}\_\tau'^{[T]}=\mathbf{x}\_\tau^{[T]}-\mathbf{x}\_r^{[T]}$ where $\mathbf{x}\_\tau^{[T]}$ are the original latent variables, the resulting trajectory $\mathbf{x}\_\tau'^{[T]}$ effectively takes the same form as DDPM (see proof of Proposition A.2 in appendix). This finding enables us to readily transform the diffusion trajectory of $\mathbf{x}\_\tau'^{[T]}$ into an SDE/ODE (see Song et al. [1]), while we only need to add $\mathbf{x}\_r^{[T]}$ back after sampling from the transformed SDE/ODE.
>
> The convenient transformation from GeoTDM to its ODE formulation enables us to leverage fast samplers such as DDIM [2]. We provide extra experiments on Nbody dataset by performing DDIM sampling using the method described above. The results with different sampling steps are depicted below.
>
> ||ADE|FDE|
> |-|-|-|
> |DDPM-1000|0.110|0.258|
> |DDIM-100|0.118|0.271|
> |DDIM-50|0.127|0.289|
>
> Interestingly, the performance degradation is marginal even when using only 50 sampling steps with DDIM sampler. GeoTDM with 50 sampling steps still outperforms the best baseline EqMotion which has an ADE/FDE of 0.141/0.310. This underscores the potential of streamlining GeoTDM for faster inference.
>
> [1] Song et al. Score-Based Generative Modeling through Stochastic Differential Equations. In ICLR'21.
>
> [2] Song et al. Denoising Diffusion Implicit Models. In ICLR'21.
>
>
> > **[Q2] Except for the part of processing temporal information, could you compare more details about the proposed EGTN/EGCL with the E(3) Equivariant Diffusion Model in the paper [1]?**
>
> Except for processing temporal information, our model has several advantages compared with EDM [1]:
>
> EDM is designed for molecule generation from scratch and does not directly support conditioning. GeoTDM can perform unconditional generation and conditional generation with the help of the proposed equivariant cross-attention that enables conditioning on a given trajectory.
>
> GeoTDM also employs a flexible learnable prior parameterized by a lightweight neural network. The prior is carefully designed such that it subsumes existing parameterizations of the prior, including the center-of-mass (CoM) based prior used in EDM (see Theorem A.4 in Appendix A.3). We have shown in Sec. 5.3 that our prior yields superior performance over CoM-based prior used in EDM.
>
> We have also systematically compared GeoTDM with other related works in Table 13 of Appendix C.6.
>
> [1] Hoogeboom, Emiel, et al. "Equivariant diffusion for molecule generation in 3d." International conference on machine learning. PMLR, 2022.
>
> > **[Q3.1] Why the geometric restrictions are also important in N-body simulation?**
>
> In N-body simulation the particles are governed by physical laws such as Columb law, Hooke's law, and Newton's law of gravitation. These physical laws ubiquitously abide by physical symmetry, which is mathematically revealed by equivariance. That is, when the system is rotated/translated, the trajectories driven by these physical laws will rotate/translate in the same way, hence motivating our geometric restrictions of GeoTDM to be SE(3)-equivariant.
>
> > **[Q3.2] What is the 'geometric structure' in these scenarios?**
>
> * In physical simulation, the geometric structure is instantiated as a fully-connected geometric graph since each pair of the particles preserves some interaction (e.g., Columb force for Charged Particles dataset).
> * For molecular dynamics, the geometric structure is defined as the molecular graph, with the node indicates the atoms and the edges corresponds to the chemical bonds.
> * For pedestrain trajectory, the geometric structure is a radius graph with node being each pedestrian and edges are connected if two pedestrians are within a certain distance.
>
> > **[Q3.3] It would be better if the author could provide more background knowledge about geometric trajectory and the physics simulation/molecular dynamics in the appendix.**
>
> Thank you for the advice! We will add one section in the appendix to introduce backgrounds on geometric trajectory, physics simulation, and MD.
>
>
> > **[Q4] Based on the visualization results in the appendix, the change in the molecular structure seems to be 'small'. Will such short trajectories be practically useful/meaningful in application?**
>
> In computational chemistry, MD sampling is usually performed over metastable molecular structures instead of highly unstable states [1], which accounts for why the changes in the visualization seem to be relatively small. However, in our MD17 experiments each trajectory has 30 frames, down-sampled from the raw data by a factor of 10, corresponding to a length of 300 in raw data which is sufficiently long to capture important chemical prior such as the rotation of the methyl group in aspirin (see Fig. 5).
>
> Moreover, in Appendix C.2 we have provided a sampling algorithm to sample much longer trajectories using our GeoTDM trained on shorter trajectories. A demonstration for a long trajectory (120 frames) obtained from our model (trained on 30 frames) can be found in `aspirin_longtraj.gif` in the supplementary file. It demonstrates our GeoTDM can produce long and stable MD trajectories, which highlights its practical significance.
>
> [1] Durrant et al. Molecular dynamics simulations and drug discovery. BMC biology. 2011.
>
> **We sincerely hope our response could address your concerns!**

---

> > ### Comment · Reviewer_RWWk · 2024-08-09
> >
> > Thank you very much for your detailed response! Most of my concerns have been addressed. Also combined with other reviewers' comments and rebuttals, I decided to raise the score.

---

> > > ### Author Response · Authors · 2024-08-09
> > > **Thank you for the feedback!**
> > >
> > > Dear Reviewer RWWk,
> > >
> > > Thank you very much for your feedback and recognition of our efforts! We greatly appreciate your valuable suggestions and will incorporate the discussions into the final version.
> > >
> > > Best,
> > >
> > > Authors

---

### Official Review · Reviewer_CUJA · 2024-07-09

**Soundness:** 3
**Presentation:** 3
**Contribution:** 2
**Rating:** 6
**Confidence:** 2

**Summary:**

In this paper, the authors propose geometric trajectory diffusion models (GeoTDM) to model temporal distribution of geometric trajectories
while keeping the desirable physical symmetry of the trajectories.

**Strengths:**

The authors impose certain constraints of SE(3)-invariant on the prior and transition kernel to keep the desirable physical symmetry of the trajectories.

**Weaknesses:**

I am concerned that dynamic system and  rotation translational invariant has already been studied in the previous research (listed below, but not quoted here), not like the authors claimed that this is the first study on this topic. There might be some overlaid here.

Both algorithms use the same E(n) equivariant graph neural networks.

Missing reference:
Fang Wu, Stan Z. Li, DIFFMD: A Geometric Diffusion Model for Molecular Dynamics Simulations. The work in this previous paper requires that the dynamics must be invariant to rotation or translation.

The paper needs more experimental results to support it.

**Questions:**

P5, line 185: In the definition of the operator P, ⊗ is not defined? It seems to me no need to define operator P if P(x) is defined.

**Limitations:**

Yes.

---

> ### Author Rebuttal · Authors · 2024-08-06
>
> Thank you for your constructive review and suggestions! We provide point-to-point response below.
>
> > **[W1] I am concerned that dynamic system and rotation translational invariant has already been studied in the previous research (listed below, but not quoted here), not like the authors claimed that this is the first study on this topic. There might be some overlaid here. Missing reference: [1]**
>
> Thank you for bringing up the related work DiffMD [1]. We will definitely cite and discuss the work [1] in the revised version.
>
> However, our work is substantially different from DiffMD in the following aspects.
>
> 1. DiffMD only models the distribution of the next frame $\mathbf{x}^{(t+1)}$ given the current frame $\mathbf{x}^{(t)}$, while our GeoTDM directly models the joint distribution of an entire trajectory $\mathbf{x}^{[T]}$ with a collection of multiple frames. Therefore, DiffMD is not explicitly a generative model on geometric trajectories, while to the best of our knowledge our GeoTDM is the first. Given such nature, DiffMD is vulnerable to error accumulation when generating trajectory with multiple frames at inference time, while our GeoTDM is able to jointly model the correlations of multiple frames within a trajectory and thus incurs much smaller error. We also empirically verify this argument in the additional experiment provided below.
> 2. DiffMD relies on taking as input a given frame, while GeoTDM can both perform unconditional generation as well as conditional generation. It is unclear how to generate a geometric trajectory from scratch using DiffMD, for example, in the unconditional generation settings in our paper (Table 4 and 5).
> 3. DiffMD is specifically designed for modeling molecular dynamics which heavily leverages particular geometric features like bond angles and dihedral angles in molecules. By contrast, our GeoTDM, without relying on domain-specific features, has been demonstrated to perform promisingly across a wide suite of benchmarks including physical simulation, molecular dynamics, and even pedestrian trajectory forecasting.
>
> We also add additional experiments perform a thorough comparison. Since there is no public code released for DiffMD, we implement a variant of DiffMD (denoted DiffMD*) which replaces the task-specific backbone proposed in DiffMD by EGNN. We adopt the original training loss and sampling procedure proposed in DiffMD. We also set the same number of layers and hidden dimension as our GeoTDM, ensuring a fair comparison. Since DiffMD is not directly applicable to unconditional generation, we benchmark it in the conditional generation setting on Nbody and MD17 datasets. The results are presented below.
>
> ||Particle|Spring|Gravity|
> |-|-|-|-|
> |DiffMD*|0.170/0.382|0.0093/0.0252|0.298/0.724|
> |GeoTDM|0.110/0.258|0.0030/0.0079|0.256/0.613|
>
> ||Asp|Ben|Eth|Mal|Nap|Sal|Tol|Ura|
> |-|-|-|-|-|-|-|-|-|
> |DiffMD*|0.152/0.301|0.025/0.051|0.193/0.406|0.209/0.477|0.090/0.142|0.131/0.306|0.129/0.235|0.161/0.260|
> |GeoTDM|0.107/0.193|0.023/0.039|0.115/0.209|0.107/0.176|0.064/0.087|0.083/0.120|0.093/0.121|0.074/0.099|
>
> It is observed that GeoTDM, by modeling the joint distribution of all frames within the geometric trajectory, consistently yields lower error than DiffMD*, which only captures the distribution of one target frame and thus needs to perform iterative roll-out for the entire trajectory, incuring extra error accumulation. We will add these discussions to the paper.
>
> [1] Fang Wu, Stan Z. Li, DIFFMD: A Geometric Diffusion Model for Molecular Dynamics Simulations.
>
>
> > **[W2] The paper needs more experimental results to support it.**
>
> In this paper, we have performed exhaustive experiments on tasks including physical simulation (3 datasets), molecular dynamics (8 datasets in MD17 as well as an additional OC22 dataset [1], see Appendix C.1), and pedestrian trajectory forecasting (5 datasets in ETH-UCY). For physical simulation and MD, we also benchmark on both unconditional and conditional generation scenarios. In Sec. 5.3 we conduct extensive ablation studies and some additional use cases including leveraging GeoTDM to perform trajectory interpolation and optimization. We also propose a sampling algorithm to sample much longer trajectories using our GeoTDM trained on shorter trajectories through model composition, and verifying its feasibility by experiment (see `aspirin_longtraj.gif` in the supplementary file). The above experiments not only cover a diverse range of benchmarking suites but also involve in-depth use cases of how to unleash the potential of GeoTDM for novel application scenarios and tasks. We are glad to include more if the reviewer has concrete suggestions on what extra experiments are needed.
>
>
> > **[Q1] P5, line 185: In the definition of the operator P, ⊗ is not defined? It seems to me no need to define operator P if P(x) is defined.**
>
> Thank you for raising this point. The symbol $\otimes$ refers to the Kronecker product. For instance, $I_D\otimes I_N=I_{DN}$ where $I_D$, $I_N$, and $I_{DN}$ are the indentity matices with shape $D\times D$, $N\times N$, and $DN\times DN$, respectively.
>
> Here $\mathbf{P}$ is the matrix representation of the linear function $P(\cdot)$, i.e., $\mathbf{P}\mathbf{x}=P(\mathbf{x})$, where the left hand side is matrix multiplication. We formally define the operator $\mathbf{P}$ here in order to clearly show that it is a linear operation with rank $(TN-1)D$, which facilitates our subsequent analysis (more details in Appendix A.1). However, we agree that it is not necessary to define $\mathbf{P}$ in the main text when $P(\cdot)$ has already been introduced, and we will move it to Appendix A.1 instead.
>
> **We sincerely hope our response could address your concerns!**

---

> > ### Comment · Reviewer_CUJA · 2024-08-09
> >
> > Thank you for your detailed response! After I read all your rebuttals and other reviewers comments. I decided to stay at the same rating.

---

> ### Author Response · Authors · 2024-08-09
> **Thank you for the feedback**
>
> Dear Reviewer CUJA,
>
> We greatly appreciate your advice and promise to incorporate the discussions in the final version. In particular, we also summarize the core distinctions between our GeoTDM and DiffMD [1] as follows:
>
> |  | Distribution Modeled |  Unconditional | Prior | Backbone |
> |-|-|-|-|-|
> | DiffMD [1]    | Frame    | $\times$ | Fixed Pointwise Prior | Specialized for molecules |
> | GeoTDM    | Trajectory  | $\checkmark$ | Learned Flexible Prior | General for $n$-D geometric data |
>
> Overall, our experiments have verified the advantages of our approach over DiffMD in terms of empirical performance. We highly appreciate it if you could kindly consider adjusting your evaluation if we have addressed your concerns. Thank you!
>
> Best,
>
> Authors
>
> [1] Fang Wu, Stan Z. Li, DIFFMD: A Geometric Diffusion Model for Molecular Dynamics Simulations.

---

> ### Comment · Reviewer_CUJA · 2024-08-10
>
> Thank you for your new update! What troubled me is that I am still not convinced what you said that: "DiffMD is not explicitly a generative model on geometric trajectories, while to the best of our knowledge our GeoTDM is the first. ". I believe that both of your methods are a generative model, but each of you approached in some detailed difference, so I need you to clarify this in more details. While you claimed that your "modeling the joint distribution of all frames within the geometric trajectory", what I saw is you still look at the individual conditional probability like DiffMD, see line 193. It seems like you take into data sampling of time into consideration, to me, I think this is considered as generalize of DiffMD. For the above reason, I believe that is why you both used Equivariant Graph Convolution Layer.

---

> > ### Author Response · Authors · 2024-08-12
> > **Follow-up on the clarifications**
> >
> > Dear Reviewer CUJA,
> >
> > Thank you for your follow-up discussion and we have clarified your questions in detail. As the discussion period is coming to an end, we greatly appreciate it if you could let us know whether the clarifications address your concern and we are willing to offer more explanations if there are any questions. Thank you!
> >
> > Best,
> >
> > Authors

---

> ### Author Response · Authors · 2024-08-10
> **Further Clarifications**
>
> Dear Reviewer CUJA,
>
> Thank you very much for the timely feedback and the follow-up question! We are glad to further clarify your concern.
>
> > What troubled me is that I am still not convinced what you said that: "DiffMD is not explicitly a generative model on geometric trajectories, while to the best of our knowledge our GeoTDM is the first. ". I believe that both of your methods are a generative model, but each of you approached in some detailed difference, so I need you to clarify this in more details.
>
> Yes, we agree that both DiffMD and GeoTDM are generative models. The core difference lies in whether the method is modeling the distribution of **geometric trajectories** $\mathbf{x}^{[T]}=[\mathbf{x}^{(0)},\mathbf{x}^{(1)},\cdots,\mathbf{x}^{(T-1)}]\in\mathbb{R}^{T\times N\times D}$ which is a **sequence/collection of frames**, or a **single** frame $\mathbf{x}^{(t)}\in\mathbb{R}^{N\times D}$. In particular, our GeoTDM explicitly models the distribution of geometric trajectories $p(\mathbf{x}^{[T]}|\cdot)$ (see line 116-122), while DiffMD is designed to model the distribution of the next frame $p(\mathbf{x}^{(t+1)}|\cdot)$ (see Model Overview section on Page 3 of DiffMD [1]), where $\cdot$ denotes certain conditioning if applicable (e.g., the previous frames).
>
> Such difference leads to several distinctions in model design, inference, and empirical performance, as we will discuss below.
> * **Model design.** In order to jointly model $p(\mathbf{x}^{[T]}|\cdot)$, in the transition kernel we need to additionally handle the temporal dimension with size $T$. This motivates us to design the equivariant temporal attention layer (Eq. 5-7), which is absent in DiffMD since DiffMD does not require to explicitly model temporal correlation. Our framework also enables conditioning on a trajectory with multiple frames while DiffMD is based on Markovian assumption and always consumes the previous one single frame to generate the next frame.
> * **Inference.** The benefit of directly modeling the distribution of geometric trajectories over single frames also presents at inference time. Within one diffusion loop, our GeoTDM can generate an entire trajectory with $T$ frames, while DiffMD requires an additional outer loop that sweeps through $T$ frames in order to achieve the same effect.
> * **Performance.** We also demonstrate the benefit in terms of empirical performance. Since DiffMD does not consider the correspondence between multiple frames, it is more vulnerable to error accumulation when generating a long trajectory through iterative rolling out. Our extra experiment provided in the rebuttal verifies this point, with GeoTDM outperforming DiffMD by a remarkable margin.
>
>
>
> > While you claimed that your "modeling the joint distribution of all frames within the geometric trajectory", what I saw is you still look at the individual conditional probability like DiffMD, see line 193. It seems like you take into data sampling of time into consideration, to me, I think this is considered as generalize of DiffMD.
>
> Thank you for the question. However, there might be a misunderstanding here and we would like to respectfully clarify.
>
> Specifically, line 193 depicts the transition kernel $p\_\theta({\mathbf{x}\_{\tau-1}^{[T]}}|\mathbf{x}\_{\tau}^{[T]})$. Here $\tau$ refers to the **diffusion step** instead of the **frame index on the geometric trajectory**. The superscript $[T]$ indicates that the latent variable here is a geometric trajectory instead of a single frame. Therefore, the individual conditional probability is enforced on **diffusion step**, which is due to the Markovian assumption of the diffusion process. This is a common practice for diffusion models and is shared in both GeoTDM and DiffMD, and we are not claiming any difference on this point.
>
> However, we are **not** enforcing conditional independence on the actual temporal dimension of the geometric trajectory, while DiffMD has a Markovian assumption on molecular dynamics and only models the distribution of a single frame. This is what we refer to by stating
>  "modeling the joint distribution of all frames within the geometric trajectory". This point leads to the core difference between these two approaches, as we have discussed.
>
>
> > For the above reason, I believe that is why you both used Equivariant Graph Convolution Layer.
>
> We are both using EGCL to process spatial information on the geometric structure. However, since we are modeling the whole trajectory, we need to additionally introduce the equivariant temporal attention layer to process the temporal correspondence, which never presents in DiffMD.
>
>
> Thank you again for the follow-up discussion! We promise to include the discussion and the distinctions between these two methods in the manuscript. Please let us know if this addresses your concern and we are happy to further clarify if you have any questions.
>
> Best,
>
> Authors

---

> ### Comment · Reviewer_CUJA · 2024-08-12
>
> Thank you for your response and clarification! I decide to raise my rating to 6: Weak Accept. I trust you can revise your paper according to the discussion.

---

> > ### Author Response · Authors · 2024-08-12
> > **Thank you for the supportive feedback!**
> >
> > Dear Reviewer CUJA,
> >
> > Thank you for the supportive feedback and the insightful comments that help us improve the manuscript! We will include these in the final version.
> >
> > Best,
> >
> > Authors

---

### Official Review · Reviewer_UDq7 · 2024-07-11

**Soundness:** 3
**Presentation:** 4
**Contribution:** 2
**Rating:** 5
**Confidence:** 4

**Summary:**

The paper introduces geometric trajectory diffusion models for generation of particle, pedestrian, or molecular trajectories. The architecture consists of EGNN layers within a temporal frame and temporal attention across frames computed with relative temporal ecodings. The architecture is shown to possess the appropriate equivariances. Key points of novelty include a learnable prior as a function of conditioning frames in the conditional generation case. Experiments are conducted on particle, molecular, and pedestrian dynamics datasets.

**Strengths:**

* The method is a timely extension and synthesis of geometric diffusion models and trajectory diffusion models.
* As demonstrated in the experiments, the method has broad applicability across many machine learning domains.
* The experiments are thorough. The ablation studies validating the somewhat unconventional choices of conditional prior are appreciated.
* The paper is clearly written and the exposition of the many experiments is handled very cleanly.

**Weaknesses:**

* The work does not score highly in conceptual novelty. While all architecture choices are sensible, they are relatively straightforward and do not seem surprising, insightful, or inspired. An extension or exploration of more sophisticated equivariant architectures could have strengthened the paper.
* The exposition of the method drags at times, re-proving well-known statements about equivariance in neural networks and geometric diffusion models.
* The experiments, while diverse and broad, also suffer from being scattered and not necessarily the most convincing individually. Little effort is spent on exploring the ways in which a trajectory diffusion could be employed and the new problems that could be solved with the mode. To me, the novel capabilities of such a method are more exciting than marginal gains across a number of established datasets, executed checkbox-style.

**Questions:**

See above

**Limitations:**

Yes

---

> ### Author Rebuttal · Authors · 2024-08-06
>
> Thank you for your constructive review and suggestions! We provide point-to-point response below.
>
> > **[W1] The work does not score highly in conceptual novelty. While all architecture choices are sensible, they are relatively straightforward and do not seem surprising, insightful, or inspired. An extension or exploration of more sophisticated equivariant architectures could have strengthened the paper.**
>
> We will first elaborate on the motivation and insights of our architectural design. We also add extra experiments that explore more sophisticated equivariant networks under the framework of our GeoTDM.
>
> **Motivation and insights of the architecture.** Our EGTN is a stack of spatial layers and our proposed equivariant attention temporal layers in an alternated fashion.
> * For spatial layer, we use the widely adopted EGNN layer, akin to existing geometric diffusion models for static structures. Notably, this module is adopted in a plug-and-play manner, which enables us to switch into other advanced backbones depending on different data and task. We also provide extra experiments by switching into more advanced equivariant networks in the part below.
> * To process temporal information, we propose a novel equivariant attention layer that involves several core designs including relative temporal embedding, satisfaction of equivariance, and extension to cross-attention for conditioning. The utilization of attention here is inspired by the success of Transformers in processing sequence data such as text and audio, since here we are handling a sequence of frames.
>
> We have also performed thorough ablation studies in Table 6, which aim to provide deeper insight about how these design choices matter.
>
> **Exploration of more sophisticated equivariant networks.** We also provide additional experiments on exploring more sophisticated equivariant networks in the framework of our GeoTDM. We replace EGCL by an advanced backbone Equiformer [1] (dubbed GeoTDM*) which enables utilizing higher order equivariant tensor. We use the codebase of [1]. The results on MD17 conditional generation are exhibited below.
>
> ||Asp|Ben|Eth|Mal|Nap|Sal|Tol|Ura|
> |-|-|-|-|-|-|-|-|-|
> |GeoTDM*|0.099/0.184|0.023/0.036|0.098/0.189|0.117/0.184|0.060/0.092|0.075/0.113|0.112/0.130|0.067/0.094|
> |GeoTDM|0.107/0.193|0.023/0.039|0.115/0.209|0.107/0.176|0.064/0.087|0.083/0.120|0.093/0.121|0.074/0.099|
>
> We observe that the performance is generally enhanced across several molecules. However, the sampling time has been dramatically increased since Equiformer is more computationally exhaustive compared with EGNN. Besides, it is a domain-specific backbone that cannot be directly applied to other tasks like pedestrian trajectory modeling, which limits the broadness of application.
>
> [1] Liao et al. Equiformer: Equivariant Graph Attention Transformer for 3D Atomistic Graphs. In ICLR'23.
>
>
> > **[W2] The exposition of the method drags at times, re-proving well-known statements about equivariance in neural networks and geometric diffusion models.**
>
> The formal statements presented in this paper are fundamentally different from existing works and are not simply re-proved counterparts. The theoretical framework initiates with the formal definition of geometric trajectory (line 116), which is distinct from the static structures discussed in existing literature, being an extension of them with an additional temporal dimension. We then rigorously defined how the group action $g\in SE(3)$ operates on geometric trajectory (line 129). Notably, Theorem 4.2-4.4 are introduced and proved under the definition of geometric trajectory and the group action enforced on it, which generalize the theorems in existing works and subsume them as special case when $T=1$. These theorems are presented for completeness and mathematical rigor of the paper.
>
> > **[W3] The experiments, while diverse and broad, also suffer from being scattered and not necessarily the most convincing individually. Little effort is spent on exploring the ways in which a trajectory diffusion could be employed and the new problems that could be solved with the mode. To me, the novel capabilities of such a method are more exciting than marginal gains across a number of established datasets, executed checkbox-style.**
>
> Thank you for recognizing the diversity and broad coverage of the experiment and the suggestions on exploring novel capabilities of GeoTDM.
>
> Indeed, in the paper we have already explored a few directions regarding the novel capabilities of GeoTDM:
>
> 1. We adopt GeoTDM to OC22 [1], a dataset of novel large-scale catalytic MD systems. The detailed experimental setup and results have been presented in Appendix C.1. The results demonstrate the potential of leveraging GeoTDM to simulate the dynamics of catalyst system, which bears significance in designing novel catalyst systems.
> 2. We also demonstrate the capability of GeoTDM in performing temporal interpolation of trajectories and trajectory optimization in Sec. 5.3. GeoTDM is capable of handling these special tasks since it enjoys the benefits of being a controllable diffusion model that captures the joint distribution of the entire trajectory, while the baselines in existing works are not. These experiments have shown the promise of employing GeoTDM to more applications such as designing chemical reaction path given the initial and target system state, which is an interesting future direction.
> 3. We propose an approach to generate long trajectory using GeoTDM trained on shorter trajectories through model composition in Appendix C.2. The demonstration reveals the capability of our model to produce long and stable MD trajectories, which also highlights a novel application scenario of GeoTDM.
>
> [1] Tran et al. The Open Catalyst 2022 (OC22) Dataset and Challenges for Oxide Electrocatalysts.
>
> **We sincerely hope our response could address your concerns!**

---

> > ### Comment · Reviewer_UDq7 · 2024-08-09
> >
> > I appreciate the detailed author response. However it has not moved the needle substantively on my concerns and I will keep the score.

---

> > > ### Author Response · Authors · 2024-08-09
> > > **Thank you for the feedback**
> > >
> > > Dear Reviewer UDq7,
> > >
> > > Thank you for the timely reply. To help substantively address your concerns, we summarize and further illustrate our response below.
> > >
> > > For **W1**, we added extra experiments that explore the way to combine our proposed equivariant temporal layer with more sophisticated equivariant architectures, e.g., Equiformer, that helps further boost the performance on specific tasks like MD. We will also include these discussions in the paper.
> > >
> > > For **W2**, we clarified the difference between theorems presented in this paper and previous work. Besides, we provided necessary derivations in Appendix A where we have interestingly found that our GeoTDM also preserves the simplified loss (Eq. 12) which is yet not straightforward. Notably, in Theorem A.4 we also theoretically justify our learnable prior subsumes existing parameterizations. All of these constitute our theoretical contributions.
> > >
> > > For **W3**, We have also presented various explorations on approaching new problem with GeoTDM, including modeling large scale catalyst dynamics (Appendix C.1), interpolating trajectories given initial and target states (Sec. 5.3), optimizing the input trajectory towards the learned distribution (Sec. 5.3), and performing generation through longer time horizon (Appendix C.2).
> > >
> > > Please let us know if you have any concerns in particular or any advice that could help further improve the paper and we are more than willing to address them. Thank you!
> > >
> > > Best,
> > >
> > > Authors

---

### Official Review · Reviewer_e5nM · 2024-07-12

**Soundness:** 3
**Presentation:** 3
**Contribution:** 3
**Rating:** 6
**Confidence:** 1

**Summary:**

The paper proposed the first diffusion model for modeling the temporal distribution of 3D geometric trajectories, while previous works only operate on static structures. It demonstrates the equivariant temporal kernels can lead to density with desired symmetry and develop a novel transition kernel leveraging SE(3)-equivariant spatial convolution and temporal attention. The experiments demonstrates it can generates realistic geometric trajectories with significantly higher quality.

**Strengths:**

1. The geometric trajectory diffusion model is novel to me.

2. The EGTN that operates on geometric trajectories permits conditioning upon a given trajectory using equivariant cross-attention

3, The experiments demonstrate that GeoTDM achieves SOTA performance on both unconditional and conditional trajectory generation tasks.

4. The work can be extended to applications like temporal interpolation and trajectory optimization

**Weaknesses:**

None

**Questions:**

None

---

> ### Author Rebuttal · Authors · 2024-08-06
>
> Thank you for your constructive review and recognition of our work! Please let us know if you have any questions and we are more than happy to answer.

---

### Decision · Program_Chairs · 2024-09-25

**Decision:**

Accept (poster)

**Comment:**

The paper introduces GeoTDM, a diffusion model designed for modeling the temporal distribution of 3D geometric trajectories, incorporating SE(3)-equivariant spatial convolution and temporal attention. Reviewers appreciated the novel approach, noting its broad applicability to physical simulations, molecular dynamics, and pedestrian motion, and recognized the well-presented methodology and extensive experiments validating its effectiveness.

Reviewers highlighted the paper’s strengths in its broad applicability across multiple domains, well-executed and thorough experiments, and the clear and detailed presentation of the methodology. The use of equivariant temporal kernels and learnable geometric priors was praised for capturing complex spatial and temporal dynamics effectively. Additionally, the ablation studies and visualizations provided valuable insights into the model’s performance, showcasing its practical utility and setting it apart from existing approaches.